# Unifying description of the damping regimes of a stochastic particle in a periodic potential

**Antonio Piscitelli[1]★ and Massimo Pica Ciamarra[1,2]†**

**1** Division of Physics and Applied Physics, School of Physical and Mathematical Sciences, Nanyang Technological University, Singapore
**2** CNR–SPIN, Dipartimento di Scienze Fisiche, Università di Napoli Federico II, I-80126, Napoli, Italy

★ antps@ntu.edu.sg,   † massimo@ntu.edu.sg

## Abstract

We analyze the classical problem of the stochastic dynamics of a particle confined in a periodic potential, through the so called Il'in and Khasminskii model, with a novel semi-analytical approach. Our approach gives access to the transient and the asymptotic dynamics in all damping regimes, which are difficult to investigate in the usual Brownian model. We show that the crossover from the overdamped to the underdamped regime is associated with the loss of a typical time scale and of a typical length scale, as signaled by the divergence of the probability distribution of a certain dynamical event. In the underdamped regime, normal diffusion coexists with a non-Gaussian displacement probability distribution for a long transient, as recently observed in a variety of different systems. We rationalize the microscopic physical processes leading to the non-Gaussian behavior, as well as the timescale to recover the Gaussian statistics. The theoretical results are supported by numerical calculations, and are compared to those obtained for the Brownian model.


## Contents



# 1  Introduction

The theoretical description of the stochastic motion of a particle confined in a periodic potential is a standard problem in statistical physics, which is relevant to a variety of different context, including superionic conductors [1], colloids in light fields [2,3], market evolution models [4], supercooled liquids [5–7], diffusion of atoms in optical lattices [8,9], diffusion of molecules at liquid/solid interfaces [10], transport in electronics [11], research strategies in biology [12]. In the absence of correlations, the central limit theorem assures that at long times the probability distribution $P(r,t)$ that a particle move of a distance $r$ in a time $t$ approaches a Gaussian,

$$P(r,t) = \frac{1}{(4\pi Dt)^{d/2}} \exp\left(-\frac{r^2}{2dDt}\right),$$

where $d$ is the spatial dimensionality and $D$ the diffusion constant. This implies that the mean square displacement (MSD) asymptotically grows linearly in time, $\langle r^2(t) \rangle = 2dDt$, a feature known as Fickian diffusion. In this line of research, the main problem is the determination of the diffusion constant as a function of the model parameters which specify the confining potential, and the features of the interaction of the diffusing particle with the heat bath. Recently, there has also been great interest in the determination of the temporal evolution of the distribution $P(r,t)$, that appears to have universal features. Specifically, a number of different systems exhibit a long transient during which the displacement distribution is not Gaussian, but the dynamics is Fickian with the mean square displacement growing linearly in time [13], a feature termed as Brownian non-Gaussian dynamics. Indeed, a Brownian non-Gaussian dynamics is observed, for instance, in dense colloidal suspensions [5,14–16], granular materials [17–22], supercooled liquids and structural glasses [7,21,23,24], gels [25], plasmas [26], biological cells [27–33], networks or active suspensions [32,34,35], turbulent flow [36] and finance [37].

The stochastic motion of a particle confined in a one-dimensional periodic potential $V(x)$ is a problem characterized by three time scales. One timescale, $\omega_b^{-1}$, originates from the curvature of the potential on the top of the barrier, $\omega_b^2 = -\frac{\partial^2 V}{\partial x^2}\big|_{\text{top}}$, and is related to the time the particle needs to cross the barrier. The other two timescales characterize the interaction with the heat bath, and measure the time the particle needs to thermalize, $t_{\text{therm}}$, and the typical time interval in between two collisions with the heat bath, $t_c$. For instance, if the interaction with the heat bath is due to the collisions of a tracer particle with bath molecules, then on increasing the density of the bath the collisional timescale decreases [38]. The ratio between the thermalization timescale and timescale fixed by the potential defines the damping regime of the dynamics, which is overdamped if $t_{\text{therm}}\omega_b \ll 1$, and underdamped if $t_{\text{therm}}\omega_b \gg 1$.

Traditionally, the stochastic motion of a particle confined in a potential is investigated assuming the collisions with the heat bath particles to occur continuously in time, $t_c = 0$. This leads to a Langevin description of the dynamics, $m\ddot{x} = -dV(x)/dx - \gamma m\dot{x} + \xi(t)$, where $\xi(t)$ is a Gaussian with noise, $\langle \xi(s)\xi(t) \rangle = 2T\gamma m\delta(t-s)$. Solutions of this equation, and of the associated Fokker-Plank equation, are only available in the overdamped and in the underdamped regime of the dynamics, where they are obtained through different approximations. Specifically, one considers that in the overdamped limit the tracer position is a slow and diffusing variable, while in the underdamped limit the energy has these features [39]. More recently it has been shown that the two limits can be accessed through a singular perturbation expansion [40], that is performed using as a small parameter $\gamma^{-1}$ in the overdamped limit, and $\gamma$ in the underdamped limit. The use of different approximations to investigate the different regimes of the dynamics makes impossible to address the feature of the dynamics as one move from one regime to the other. Similarly, to obtain the precise shape of $P(r,t)$ at all the times, including on the tails, would require the exact solution of the Fokker-Planck equation, for the given potential and initial condition, which is in general not known.

In this paper, we investigate the diffusion and the evolution of the displacement probability distribution of a stochastic model describing the motion of a tracer particle in a potential, which is different but closely related to the usual Brownian motion model. Specifically, we focus on the Il'in Khasminskii (IK) model [41], in which instantaneous interaction with the heat bath occurs at a constant rate $t_c^{-1}$. The interaction randomizes the particle's velocity according to the Maxwell distribution so that $t_c$ also fixes the thermalization timescale. In between two interactions with the heat bath, the tracer particle moves in the potential according to Newton's law. We develop a theoretical framework to investigate the dynamics of this model in all damping regimes. Specifically, this theoretical framework is based on the notion of 'flights', that has no analogous in the BM dynamics. A 'flight' is defined as the smooth and deterministic trajectories of a particle in between two successive interactions with the heat bath. We define flights that connect different potential wells as 'external flights', and the other as 'internal flights', as illustrated in Fig. 1. This framework allows us to address, for the first time, what features of the dynamics change as the system transient from the overdamped to the underdamped regime. We will show that, in the overdamped limit, $\omega_b t_c \ll 1$, the diffusion coefficient is set by the internal flights, while in the underdamped regime, $\omega_b t_c \gg 1$, it is set by the external flights. The crossover between the overdamped and the underdamped regimes comes with a singularity of the probability density of a particular class of flights. Physically, this divergence implies that on moving from the overdamped to the underdamped regime the dynamics loses a characteristic length scale and a characteristic energy scale.

The introduced theoretical framework does also allow for a detailed investigation of the Brownian non-Gaussian feature of the dynamics. This is of particular interest as there are not simple analytically solvable models where the Brownian non-Gaussian dynamics emerges from the solution of a physically motivated equation of motion. Indeed, current statistical models exhibiting such a dynamics are formulated at a coarse-grained level. For instance,

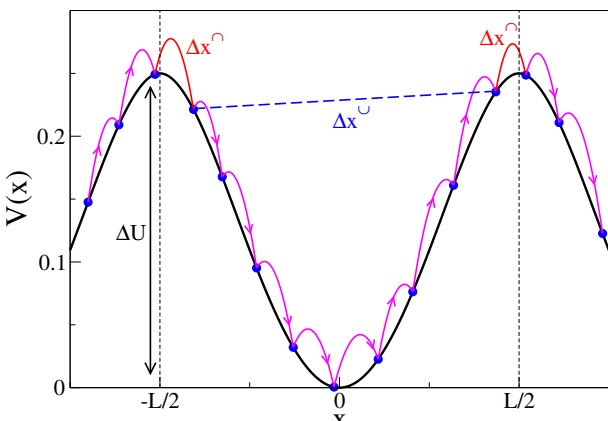

Figure 1: Schematic representation of the dynamics of Il'in Khasminskii (IK). Each arc represents the deterministic trajectory in between two successive collisions with the heat bath. External flights (red) connect different potential wells, while internal flights (blue) connects two successive external flights. The curvatures of the potential at the top and at the bottom of the barrier are respectively $\omega_b^2 = -\frac{\partial^2 V}{\partial x^2}\mid_{x=L/2}$ and $\omega_0^2 = \frac{\partial^2 V}{\partial x^2}\mid_{x=0}$. Adapted from [42].

the superstatistical approach elegantly recovers a Brownian non-Gaussian dynamics assuming the existence of an exponential distribution of diffusion constants, whose physical origin is however difficult to rationalize [13,43,44]. Alternatively, a Brownian non-Gaussian dynamics can be recovered assuming the diffusion coefficient to be itself a diffusing variable [27,45,46], as one might expect to occur for particles moving in evolving environments. Here we show that the Brownian non-Gaussian dynamics emerges as the dynamics is naturally described as the superposition of two stochastic processes, corresponding to the 'internal' flights within a well, and to the 'external' flights connecting different wells. In the underdamped regime, both 'external' and 'internal' flights affect the displacement distribution function, which leads to a heterogeneous dynamics with Brownian non-Gaussian features. We investigate the approach to the regime of normal diffusion considering the time evolution of the non Gaussian parameter (ngp) of the displacement probability distribution, which we find to scale as ngp $\propto \Gamma^{-1}t^{-1}$, where the proportionality constant is related to the ngp of the single flight length distribution, and $\Gamma^{-1}$ is the inverse rate of escape of a particle from the well.

In the following, we first define the model we have investigated, and then describe in Section 3 the theoretical approach we have developed, illustrating how this allows for the estimation of the diffusivity. In particular, in this section we provide a schematic description of the characteristic flights of the IK model along a typical trajectory and discuss the relevant couplings between the shape of the potential and the other parameters of the model in the two dynamical regimes. Section 4 describes the feature of the transition from the overdamped to the underdamped dynamics in the IK model, while the Brownian non-Gaussian features of the IK and of BM models, and the timescales associated with the recovery of the normal diffusion, are discussed in Sec. 5. A brief summary and some remarks conclude the paper.

## 2 Models

We consider two models describing the dynamics of a thermal particle confined in a one-dimensional potential, the Brownian Motion model and the Il'in Khasminskii one. While the theoretical framework we will discuss is easily generalized to generic potentials, we focus on

a periodic potential of period $L$, that in the range $-L/2 < x \leqslant L/2$ is defined as:

$$V(x) = \frac{1}{2}m\omega_0^2 x^2 - \frac{m\omega_0^2}{L^2}x^4. \tag{1}$$

We use as independent parameters the height of the potential barrier $\Delta U$ and the period $L$. The frequencies related to the curvatures at the bottom and at the top of the potential well are $\omega_0^2 = \frac{16\Delta U}{mL^2}$ and $\omega_b^2 = 2\omega_0^2$, respectively.

In the BM model, the diffusive properties of a particle confined in a one-dimensional periodic potential have been investigated in the Langevin formalisms [39,40], in which the equation of motion includes the interaction with a thermal bath at temperature $T$ and with an environment which damps the motion with a viscous friction of coefficient $\gamma$. The interaction with the thermal bath is continuous and the damping timescale is set by $\gamma^{-1}$. The overdamped and the underdamped regimes correspond, respectively, to the limits $\gamma^{-1}\omega_b \ll 1$ and $\gamma^{-1}\omega_b \gg 1$. In this paper, we will describe previous results concerning the diffusive properties of this model, and discuss new findings as concern the evolution of the displacement distribution.

In the original investigation of the IK model [41] a test particle of mass $m$, confined in a potential, elastically interacts with a heat bath particle of mass $M$. The time interval between two successive interactions is distributed like $P(\Delta t) = e^{-\frac{\Delta t}{t_c}}/t_c$. The case of collision events equispaced in time has also been treated elsewhere [48]. The interaction with the heat bath, in equilibrium at temperature $T$, is instantaneous and randomizes the velocity. In between collisions the system moves deterministically according to Newton's equation. Being $M$ the mass of the test particle and $m$ the mass of the bath particle, the reduced masses can be defined as $\mu_1 = \frac{M-m}{M+m}$ and $\mu_2 = \frac{2M}{M+m}$. Then, if $x$ and $p$ are the position and the momentum of the test particle in 1-dim, $F(x) = -\frac{\partial V(x)}{\partial x}$ is the force on the test particle due to the potential and $u(x,p,t)$ is the probability distribution of finding the particle in position $x$ with momentum $p$, the integro-differential equation that Il'in and Kashminskii found for $u$ is:

$$\frac{\partial u(x,p,t)}{\partial t} = -\frac{p}{M}\frac{\partial u(x,p,t)}{\partial x} - F(x)\frac{\partial u(x,p,t)}{\partial p}$$
$$-\frac{1}{t_c}\int_{-\infty}^{\infty} dp' P(p')\left[u(x,p,t) - u(x,\frac{p-\mu_2 p'}{\mu_1},t)\right]. \tag{2}$$

In this paper we consider the special case of "strong collision", that is $M = m$, for which the integro-differential equation for $u$ reads:

$$\frac{\partial u(x,p,t)}{\partial t} = -\frac{p}{m}\frac{\partial u(x,p,t)}{\partial x} - F(x)\frac{\partial u(x,p,t)}{\partial p} - \frac{1}{t_c}u(x,p,t) + \frac{1}{t_c\sqrt{2\pi mT}}e^{-\frac{p^2}{2mT}}\overline{u}(x,t), \tag{3}$$

where $\overline{u}(x,t)$ is the marginal distribution of the position of the particle. Eq. 2 cannot be solved in general. Although it is not formally derived from a projection procedure like the Fokker Planck, it reduces to the Fokker Planck in the limit of high collision rate and massive tracer. As a matter of fact it is more general, including physical cases where the rare and strong fluctuations prevent the diffusion limit from being attained, like in a real gas at low pressure. We stress that while in the Brownian Motion case, where $M \gg m$, the time correlation is due to the inertia of the test particle, in the case $M = m$ the time correlation $t_c$ should be thought of as the time of the mean free path of the molecules, that can be related to the inverse of the pressure of the gas [39]. Subsequent works [47,48] solved the IK dynamics for a free particle and for a particle confined in a harmonic potential, following a path integral approach, showing remarkable deviations from the Brownian Motion in the case $m = M$. We have recently investigated the diffusive properties of the IK model [42] in a periodic potential,

introducing a novel physically motivated approach that allows to obtain exact results without solving Eq. 3. We will shortly review these previous results in Section 3, providing more details which are instrumental to the study of the Brownian non-Gaussian dynamics.

The analytical solutions of the dynamics of both the BM and the IK model are validated against numerical simulations. In particular, the simulations of the IK model are carried out using an explicit Euler-Maruyama scheme to integrate the equation of motion in between two interactions with the heat bath. When a particle interacts with the heat bath, its velocity is resampled from the Boltzmann distribution $P(v) = e^{-\frac{mv^2}{2T}}/\sqrt{2\pi T/m}$. This distribution represents the invariant distribution of the dynamics and is preserved along the piecewise deterministic trajectory. In the underdamped regime, these simulations are speed-up through the use of analytical results [42] for the time needed by a particle moving deterministically to traverse a well, or to perform one oscillation inside a well.

## 3 Diffusion constant

### 3.1 Overview

We have recently investigated [42] the diffusion coefficient of the IK model, and compared it to that of the BM, in both the overdamped and the underdamped limit. The main result is that the diffusivities of the two models coincide in the overdamped regime, while they are qualitatively different in the underdamped regime. Specifically, in the overdamped limit, we have found

$$D_{\text{over}}^{\text{IK}} = e^{-\frac{\Delta U}{T}} \frac{\omega_0 \omega_b}{2\pi} t_c L^2, \tag{4}$$

which is exactly the diffusion coefficient of a BM in a periodic potential, with $t_c = \gamma^{-1}$ [49]. Conversely, in the underdamped limit the diffusion coefficient of the IK model is

$$D_{\text{under}}^{\text{IK}} \propto \frac{\Delta U}{m} t_c e^{-\frac{\Delta U}{T}}, \tag{5}$$

where the proportionality constant is weakly affected by the potential, and by the temperature. This is different from the diffusivity [40, 42] of the BM, which is given by

$$D_{\text{under}}^{\text{BM}} \propto T \gamma^{-1} e^{-\frac{\Delta U}{T}}. \tag{6}$$

In particular, the energy scale controlling the diffusivity of the IK model in the underdamped limit is the barrier height, $\Delta U$, while that controlling the diffusivity of the BM is the temperature.

In the following, we shortly review the theoretical approach we have introduced to determine the diffusion coefficient, both to provide novel physical insights into the crossover from the overdamped to the underdamped regime, as well as to establish the formalism we will use to investigate the Brownian non-Gaussian features of the dynamics.

### 3.2 Theoretical approach

Instead of searching for the full solution of the dynamics starting from Eq. 3, we follow a different approach that leads to approximate expressions in the limiting damping regimes and provides some physical insight in the crossover region as well. We determine the statistical features of the IK model in a periodic potential decomposing its dynamics into the superposition of two stochastic processes. One stochastic process describes the motion within a given potential well, while the other process describes the transition between different wells. We define as flight the trajectory of a particle in between two consecutive interactions with the heat

bath, and consider at a coarse-graining level a particle trajectory as an alternate sequence of external and of internal flights, as shown in Fig. 1. The $i$-th external flight, with length $\Delta x_i^\cap$, is a barrier crossing flight, i.e. the trajectory between the coordinates of two successive collisions happening in different wells. The $i$-th effective internal flight, with length $\Delta x_i^\cup$, is the trajectory connecting the ending point of the $i$-th external flight and the starting point of the $(i+1)$-th external flight. While an external flight involves one single interaction with the heat bath in a given potential well, an effective internal flight might involve many interactions within the same potential well, and might, therefore, consist of many flights. To simplify the notation, in the following we will refer to the effective internal flights as internal flights. The displacement of a particle at time $t$, when the particle has performed $N(t) \simeq t/t_c$ flights, can be decomposed in a contribution from the internal flights and in a contribution from the external flights. The decomposition is carried out considering that the number of external flights is $P_\cap N$, where $P_\cap$ is the probability that a thermalized particle performs a barrier crossing flight so that $R_N = \sum_i^{NP_\cap}(\Delta x_i^\cap + \Delta x_i^\cup)$. Consequently, the diffusion constant is

$$D = \lim_{t\to\infty}\frac{1}{2Nt_c}\left[\left(\sum_{j=1}^{NP_\cap}\Delta x_j^\cap\right)^2 + \left(\sum_{j=1}^{NP_\cap}\Delta x_j^\cup\right)^2\right] = D^\cap + D^\cup, \tag{7}$$

where the cross product term vanishes for symmetry reasons.

The evaluation of the diffusion coefficient through Eq. 7 involves that of the sums of variables that are, in principle, correlated. In particular, in the overdamped limit flights are short, and a barrier crossing flight will end very close to the maximum of the energy barrier separating two wells. Therefore, this flight is most probably followed by flights bringing back the particle in the starting well, rather than by flights driving the particle in the arrival well. This makes most of the $NP_\cap$ barrier crossings terms in Eq. 7 correlated. These correlations can be formally taken into account investigating the fraction $P_d$ of crossings flights that are followed by decorrelation in the arrival well, without any further recrossing. Considering that the contribution of correlated barrier crossing flights to the diffusivity vanishes, the diffusion coefficient can be expressed as

$$D = \frac{1}{2t_c}P_d P_\cap\left[\langle(\Delta x^\cap)^2\rangle + \langle(\Delta x^\cup)^2\rangle\right] = D^\cap + D^\cup, \tag{8}$$

where $\langle\cdot\rangle$ indicates the *average over uncorrelated* flights. The diffusion coefficient can be estimated evaluating the different terms that appear in Eq. 8, which is valid for every $t_c$. We describe below how this estimation can be carried out, adding a subscript $t_c \to 0$ or $t_c \to \infty$ to the different evaluated quantities to indicate respectively their overdamped and underdamped limits.

### 3.3 Barrier crossing probability, $P_\cap$

The probability that a flight crosses an energy barrier can be calculated, without loss of generality, as the probability that the coordinate of the starting point of the flight is $|x_s| < L/2$, while the coordinate of the arrival point is $|x_e| > L/2$. To evaluate this probability, we consider that a flight is characterized by three independent variables, which could be $x_s$, $x_e$ and the time of flight $\Delta t$, or equivalently $x_s$, $x_e$ and the energy $E$. Note that particles able to perform a barrier crossing flight must have an energy larger than the barrier height $\Delta U$, i.e. a positive excess

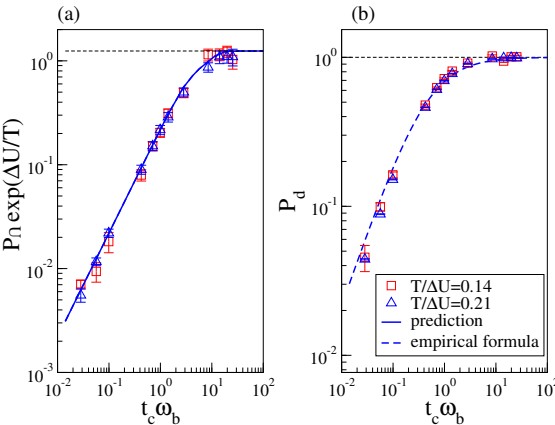

Figure 2: Panel a illustrates the $t_c \omega_b$ dependence of the probability $P_\cap$ that a flight crosses an energy barrier, normalized with the Arrhenius factor. $P_\cap$ is the long time limit of the ratio $M(t)/N(t)$, where $N(t)$ is the total number of flights until time $t$, and $M(t)$ is the number of barrier crossing flights until time $t$. The full line represents Eq. 9, the dashed line the asymptotic value for $T/\Delta U = 0.21$. Panel b shows the fraction of uncorrelated barrier crossing flights, $P_d$, described in the text. The full line is an empirical fitting formula based on the predicted behavior in the $t_c \to 0, \infty$ limits. Figure from [42].

energy $\epsilon = E - \Delta U$. Using as variables $(x_s, x_e, E)$, the barrier crossing probability results

$$
\begin{aligned}
P_\cap &= \langle \theta(|x_e| - L/2) \rangle_f \\
&= 2 \int_{-L/2}^{L/2} dx_s \int_{x_s}^{\infty} dx_e \int_{0}^{\infty} dE f(x_s, x_e, E) \theta(x_e - \frac{L}{2}) \\
&= 2 \int_{-L/2}^{L/2} dx_s \int_{L/2}^{\infty} dx_e \int_{0}^{\infty} dE f(x_s, x_e, E).
\end{aligned} \tag{9}
$$

Here

$$
f = \frac{1}{Z(T) v(x_s, E) v(x_e, E)} \frac{e^{-\frac{t_E(x_s \to x_e)}{t_c}}}{t_c} \frac{e^{-\frac{E}{T}}}{\sqrt{2\pi mT}} \tag{10}
$$

is the probability that the particle interacts with the heat bath when in position $x_s$, that through this interaction it acquires a total energy $E$, and that its flight time equals the time $t_E$ needed to travel from $x_s$ to $x_e$ with total energy $E$, which is given by

$$
t_E(x_s \to x_e) = \int_{x_s}^{x_e} \frac{dz}{v(z, E)}. \tag{11}
$$

$Z = \int_{-L/2}^{L/2} e^{-\frac{V(u)}{T}} du$ is a temperature dependent normalization constant. In Eq. 10 $v(x_s, E)$ and $v(x_e, E)$ are the velocities of the particle in the initial and in the final position, respectively. Eq. 10 is the equilibrium measure over the flights and is valid for every $t_c$. Fig. 2a shows that the theoretical prediction for $P_\cap$ of Eq. 9 compares well with the numerical results, where $P_\cap$ is evaluated as the number of barrier crossing flights over the total number of flights. The $t_c$ dependence of the barrier crossing probability is rationalized considering the properties of the flights in the overdamped and in the underdamped limit. We will show in Sec. 3.5 that in the overdamped limit $P_\cap = (\omega_0/\pi) t_c \exp(-\Delta U/T)$, and in Sec. 3.6 that in the underdamped limit $P_\cap \propto \exp(-\Delta U/T)$.

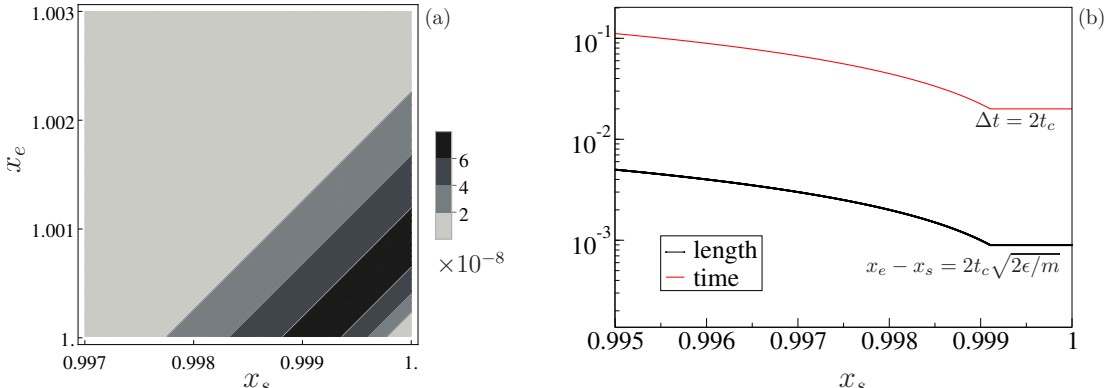

Figure 3: a) Contour plot of the contribution to the mean square displacement of the external flights, in the overdamped limit. Data are obtained at fixed excess energy $\epsilon$, and are shown as a function of $x_s < L/2 = 1$, the starting point of the external flight and of $x_e > L/2 = 1$, the endpoint of the flight. $L/2$ is the coordinate of the top of the potential barrier. Maxima are located along a straight line at 45 deg, that identifies flights of constant length at the turn of the barrier. b) Flight duration $\Delta t$ and flight length $x_e - x_s$, as a function of $x_s$, of the external flights mostly contributing to the mean square displacement, at a fixed excess energy $\epsilon$. $\Delta t$ and $x_e - x_s$ become constant as $x_s$ approaches $L/2$. For both panels, $L = 2$, $\Delta U = 1/4$, $T/\Delta U = 0.1$, $t_c = 10^{-2}$ and $\epsilon = 10^{-3}$.

## 3.4 Correlation between barrier crossing flights, $P_{\mathrm{d}}$

Subsequent barrier crossing flights are expected to be anticorrelated, in the overdamped limit. In this limit, in fact, flights are short so that barrier crossing flights will both start and end close to the top of an energy barrier. One thus expect the occurrence of many barrier crossing flights before the traces diffuses into the arrival well. The fraction $P_{\mathrm{d}}$ of crossings flights that are followed by decorrelation in the arrival well, without any further recrossing, is numerically estimated measuring the fraction $f(t)/(N(t)P_{\cap})$ of barrier crossing flights having the same direction of their predecessor. These consecutive external flights are uncorrelated, as correlated flights have opposite directions as being related to a recrossing event. Since the probability that two uncorrelated consecutive barrier crossing events have the same direction is $1/2$, we conclude that $P_{\mathrm{d}} = 2f(t)/(N(t)P_{\cap})$. Numerical results for $P_{\mathrm{d}}$ are shown in Fig. 2b. In the underdamped limit barrier crossing flights are long, and thus uncorrelated so that $P_{\mathrm{d}} = 1$. In the overdamped limit, instead, we find $P_{\mathrm{d}} \simeq 2\omega_b t_c$ as discussed in Sec. 3.5. The full line of Fig. 2b is an empirical fitting formula reproducing the asymptotic behaviors.

## 3.5 Overdamped limit

### 3.5.1 Overdamped external flights, $\langle (\Delta x^{\cap})^2 \rangle_{t_c \to 0}$

In the overdamped limit all flights are small, and do not involve large changes in the potential energy. In particular, the short barrier crossing flights experience an approximately flat potential, and will thus resemble free flights. We first show numerically that this is the case, investigating the contribution of the external flights to the MSD,

$$g_{\text{ext}} = f_\epsilon(x_s, x_e)(x_e - x_s)^2 \theta(x_e - L/2), \tag{12}$$

where $\epsilon = E - \Delta U \ll \Delta U$ is fixed. This is plotted in Fig. 3a for $\Delta U = 1/4$ and $L = 2$, as a function of $x_s < L/2$ and $x_e > L/2$. The length of the flights is $l = (x_e - 1) + (1 - x_s)$. The figure clarifies that the only not negligible contribution to the diffusion coefficient comes from

short flights, and that this contribution is maximal along the hypotenuse of an isosceles right triangle. Accordingly, the flights mostly contributing to the diffusion have different starting points but a constant length, as expected for the flights of a free particle with fixed energy and fixed flight duration. The direct measure of the length $l = x_e - x_s$ and of the duration $\Delta t$ of the barrier crossing flights confirms that these behave as free flights when $x_s$ is close to the top of the barrier, as illustrated in Fig. 3b. The squared flight length is $l^2 = v_s^2 \Delta t^2$, with velocity $v_s = \sqrt{2\epsilon/m}$, and the flight duration is $\Delta t = 2t_c$. Thus, in the overdamped regime the external flight have a well defined energy scale.

Formally, $\langle (\Delta x^\cap)^2 \rangle_{t_c \to 0}$ is evaluated as the average of the squared length of the flights crossing the top of the potential barrier over the equilibrium measure, Eq. 10, assuming the flights to be free. Using the variables $(x_s, v_s, \Delta t)$ to describe a flight, we find

$$
\langle (x_s - x_e)^2 \rangle \simeq \left( \int_{-\infty}^{0} dx_s \int_{0}^{\infty} dv_s \int_{0}^{\infty} d\Delta t \, \frac{e^{-mv_s^2/2T}}{\sqrt{2\pi T/m}} \frac{e^{-\Delta t/t_c}}{t_c} v_s^2 \Delta t^2 \theta(x_s + v_s \Delta t) \right) \times
$$
$$
\left( \int_{-\infty}^{0} dx_s \int_{0}^{\infty} dv_s \int_{0}^{\infty} d\Delta t \, \frac{e^{-mv_s^2/2T}}{\sqrt{2\pi T/m}} \frac{e^{-\Delta t/t_c}}{t_c} \theta(x_s + v_s \Delta t) \right)^{-1}
$$
$$
= \frac{12 t_c^3 T m^{1/2} \sqrt{2/\pi}}{t_c m^{3/2} \sqrt{2/\pi}} = \frac{12 T t_c^2}{m}. \quad (13)
$$

We stress that this result is valid for low $t_c$, where the free particle approximation $l^2 = v_s^2 \Delta t^2$ can be used. The term $\theta(x_s + v_s \Delta t)$ in Eq. 13 assures that we are averaging over free flights that cross the energy barrier. Without this term Eq. 13 yields the averaged squared length of all the free flights, which is $2T t_c^2/m$. This result is in agreement with the numerical calculation of the integral of Eq. 12 in the overdamped limit.

To evaluate the diffusion coefficient, we also need to calculate the barrier crossing probability $P_\cap$, Eq. 9. In the overdamped and low temperature limit this can be evaluated using the following approximations: $Z(T) \simeq \frac{\sqrt{2\pi T/m}}{\omega_0}$, $V(x) \simeq \Delta U - \frac{m\omega_b^2 (L/2 - x)^2}{2}$, and $v_x(\epsilon) \simeq \sqrt{2\epsilon/m + \omega_b^2 (L/2 - x)^2}$. As a result, we determine $P_\cap \simeq \pi^{-1} e^{-\frac{\Delta U}{T}} \omega_0 t_c$, that correctly describes the overdamped limit of Fig. 2a.

The last quantity needed to evaluate $D_{t_c \to \infty}^\cap$ is the barrier crossing probability $P_d$. From dimensional arguments it can be argued that $P_d$ should be proportional to $\omega_b t_c$, and from the simulations the constant turns out to be 2. This value of the constant makes the escape rate of the IK model equal to that of the BM, in the low temperature limit [39, 42].

Combining these results we finally estimate from Eq. 8

$$
D_{t_c \to 0}^\cap = \frac{1}{2t_c} \frac{e^{-\frac{\Delta U}{T}} \omega_0 t_c}{\pi} 2\omega_b t_c 12 T t_c^2 = e^{-\frac{\Delta U}{T}} \frac{\omega_0 \omega_b}{\pi} \frac{12 T t_c^3}{m}. \quad (14)
$$

In the last equation the relevant characteristics of the potential are encoded in the curvatures at the bottom of the well potential $\omega_0^2$ and at the top of the barrier $\omega_b^2$. In terms of the variables $\Delta U$ and $L^2$, the diffusion coefficient can be expressed as

$$
D_{t_c \to 0}^\cap \propto \frac{\Delta U}{L^2} T t_c^3 e^{-\frac{\Delta U}{T}}, \quad (15)
$$

where for the considered potential the constant of proportionality is $16/(\sqrt{2}\pi)$.

### 3.5.2 Overdamped internal flights $\langle (\Delta x^\cup)^2 \rangle_{t_c \to 0}$

The features of the internal flights are easily determined considering that each internal flight connects two barrier–crossing flights, as in Fig. 1, that are extremely short. Hence, if an internal flight connects two successive external flights having the same direction, then its length is

that of the period of the potential, while if connects successive external flights with opposite directions, then its length is negligible. If the connected external flights are uncorrelated, these two scenarios are equally likely, so that we estimate $\langle (\Delta x^{\cup})^2 \rangle_{t_c \to 0} \simeq L^2/2$. Thus, the diffusion coefficient of the internal flights results

$$D_{t_c \to 0}^{\cup} = e^{-\frac{\Delta U}{T}} \frac{\omega_0 \omega_b}{2\pi} t_c L^2. \tag{16}$$

In terms of $\Delta U$ and $L$, we obtain the following scaling:

$$D_{t_c \to 0}^{\cup} \propto \Delta U t_c e^{-\frac{\Delta U}{T}}, \tag{17}$$

where for the considered potential the constant of proportionality is $192/\pi$.

### 3.5.3 Overdamped diffusion coefficient

In the overdamped limit, $\omega_b t_c \ll 1$, the contribution $D_{t_c \to 0}^{\cap}$ to the diffusion coefficient of external flights, Eq. 15, results negligible with respect to that of the internal flights, $D_{t_c \to 0}^{\cup}$. Thus, the overall diffusion coefficient is $D_{t_c \to 0} \simeq D_{t_c \to 0}^{\cup}$, and it is given by Eq. 16. Note that Eq. 16 is exactly the diffusion coefficient of a BM in a periodic potential, with $t_c = \gamma^{-1}$ [49]. Thus, in the overdamped limit, the diffusivities of the two models are equal. We will show in Sec. 5.1 that in this limit the two dynamics have also the same MSD, as well as an almost identical displacement probability distribution, at all times.

## 3.6 Underdamped limit

In this section, we consider the internal and the external flights in the underdamped limit and determine how their diffusion constants scales with $\Delta U$ and $L$, emphasizing their weak dependence on the temperature and on the shape of the potential. The overall diffusion coefficient results dominated by the external flights, that we will, therefore, be discussed in greater detail.

### 3.6.1 Underdamped external flights

In the underdamped limit, a particle able to escape an energy barrier is expected to traverse many wells. Under the assumption that the external flights are uncorrelated and that they start from the equilibrium distribution, which we will discuss in Appendix A, the diffusion constant can be estimated as the average over the equilibrium distribution of the squared length of the barrier crossing flights $D_{t_c \to \infty}^{\cap} = \frac{1}{t_c} \langle v^2(\epsilon) \Delta t^2 \theta(x_e - L/2) \rangle_f$. Here $v$ is the average velocity of the flying particle, one can evaluate as $v(\epsilon) = L/t_b(\epsilon)$, where $t_b(\epsilon)$ is the time a particle with excess energy $\epsilon$ needs to traverse a well,

$$t_b(\epsilon) = \int_{-L/2}^{L/2} \frac{dx}{\sqrt{\frac{2}{m}(\Delta U + \epsilon - V(x))}} = \frac{L}{\sqrt{2\Delta U/m}} \tau(\zeta_\epsilon), \tag{18}$$

where $\tau(\zeta_\epsilon) = \int_{-1/2}^{1/2} \frac{dy}{\sqrt{1 + \zeta_\epsilon - 8y^2 + 16y^4}}$ and $\zeta_\epsilon = \epsilon/\Delta U$. Using as independent variables for a flight its starting position $x_s$, the initial velocity $v_s$, and the flight duration, $\Delta t$, we find

$$D_{t_c \to \infty}^{\cap} = \frac{e^{-\Delta U/T}}{t_c \sqrt{2\pi T} Z(T)} \int_0^\infty e^{-\frac{\epsilon}{T}} \left( \frac{L}{t_b(\epsilon)} \right)^2 d\frac{\epsilon}{m} \times$$

$$\int_{-L/2}^{L/2} \frac{dx_s}{v_s(\epsilon)} \left[ \int_{t_E(x_s \to L/2)}^\infty \frac{\Delta t^2 e^{-\Delta t/t_c}}{t_c} d(\Delta t) \right], \tag{19}$$

where $t_E(x_s \to L/2)$ is the time a particle with energy $E = \Delta U + \epsilon$ needs to travel from $x_s$ to the top of the energy barrier, Eq. 11. It is easy to check that the above integral is dominated by values of $\epsilon$ where $t_E(x_s \to L/2) \ll t_c$. As a consequence, we can set $t_E(x_s \to L/2) = 0$, obtaining

$$D^{\cap}_{t_c \to \infty} = \frac{2L^2 t_c e^{-\frac{\Delta U}{T}}}{\sqrt{2\pi T} Z(T)} \int_0^\infty \frac{e^{-\frac{\epsilon}{T}}}{t_b(\epsilon)} d\epsilon, \tag{20}$$

which at low temperature becomes:

$$D^{\cap}_{t_c \to \infty} = \frac{\Delta U}{m} t_c e^{-\frac{\Delta U}{T}} \left( \frac{16}{\zeta_T} \int_0^\infty \frac{e^{-\zeta_\epsilon/\zeta_T}}{\tau(\zeta_\epsilon)} d\zeta_\epsilon \right), \tag{21}$$

where $\zeta_T = T/\Delta U$. Eq. 20 is valid at all temperatures. Note that in the last expressions the precise functional form of the potential only affects the diffusivity through $\tau_\epsilon$. In the low temperature limit the term in parenthesis in Eq. 21 results only weakly temperature dependent, so that the relevant energy scale controlling the diffusion is $\Delta U$. This is at odds with what happens in the BM, where in the underdamped limit the diffusivity scales as $D \propto T\gamma^{-1} e^{-\frac{\Delta U}{T}}$, so that the relevant energy scales is the temperature [40, 42].

A better physical understanding of the underlying physical process is obtained recasting the diffusivity as $D^{\cap}_{t_c \to \infty} = \Gamma_{t_c \to \infty} \langle \lambda^2 \rangle_{t_c \to \infty}$, where $\Gamma_{t_c \to \infty}$ is the escape rate from a well, and $\langle \lambda^2 \rangle_{t_c \to \infty}$ the average square length of a barrier crossing flight. The rate of escape is evaluated as $\Gamma_{t_c \to \infty} = \lim_{t \to \infty} M_{t_c \to \infty}(t)/2t = P_{\cap, t_c \to \infty}/2t_c$ where $M(t)$ is the number of barrier crossings in both directions in a time interval $t$. Using $(x_s, v_s, t)$ as independent variables of the flight, after integrating over the time the barrier crossing probability results

$$P_{\cap, t_c \to \infty} = \frac{2e^{-\Delta U/T}}{\sqrt{2\pi T/m} Z(T)} \int_0^\infty e^{-\epsilon/T} t_b(\epsilon) d\frac{\epsilon}{m} \tag{22}$$

so that the transition rate is

$$\Gamma_{t_c \to \infty} = \frac{e^{-\frac{\Delta U}{T}}}{t_c} \left( \frac{4}{\zeta_T} \int_0^\infty e^{-\frac{\zeta_\epsilon}{\zeta_T}} \tau(\zeta_\epsilon) d\zeta_\epsilon \right). \tag{23}$$

We note that the quantity in round brackets is only weakly dependent on $T$ and $\Delta U$.

The average squared length can be estimated in two ways. On the one side, given our previous result for $D^{\cap}_{t_c \to \infty}$ and for $\Gamma_{t_c \to \infty}$, we have

$$\langle \lambda^2 \rangle_{t_c \to \infty} = \frac{D^{\cap}_{t_c \to \infty}}{\Gamma_{t_c \to \infty}} = 4t_c^2 \frac{\Delta U}{m} \left( \int_0^\infty \frac{e^{-\frac{\zeta_\epsilon}{\zeta_T}}}{\tau(\zeta_\epsilon)} d\zeta_\epsilon \right) \left( \int_0^\infty e^{-\frac{\zeta_\epsilon}{\zeta_T}} \tau(\zeta_\epsilon) d\zeta_\epsilon \right)^{-1}. \tag{24}$$

On the other side, we can also estimate $\langle \lambda^2 \rangle_{t_c \to \infty}$ as the second moment of the distribution of the length of the external flights, $P^{(F)}(l)$. This distribution can be obtained integrating $\delta\left( l - t\frac{L}{t_b(\epsilon)} \right)$ over the equilibrium measure of the flights. We obtain

$$P^{(F)}_{T, t_c \to \infty}(l) \simeq \int_{-L/2}^{L/2} dx_s \int_{t_E(x_s \to L/2)}^\infty dt \int_0^\infty d\frac{\epsilon}{m}$$
$$\frac{2e^{-\frac{\Delta U}{T}}}{\sqrt{2\pi T/m} Z(T) t_c} \frac{e^{-\frac{\epsilon}{T} - \frac{t}{t_c}}}{\sqrt{\frac{2}{m}(\Delta U + \epsilon - V(x_s))}} \delta\left( l - t\frac{L}{t_b(\epsilon)} \right) =$$
$$\left( \int_0^\infty e^{-\frac{\epsilon}{T}} e^{-\frac{l\, t_b(\epsilon)}{L t_c}} t_b^2(\epsilon) d\epsilon \right) \left( L t_c \int_0^\infty e^{-\frac{\epsilon}{T}} t_b(\epsilon) d\epsilon \right)^{-1}, \tag{25}$$

where the relation $\delta(g(t)) = \sum_i \frac{\delta(t-t_i)}{|g'(t_i)|}$ has been used, and we approximated $t_E(x_s \to L/2) = 0$ as in the derivation of Eq. 21. The numerical study of this distribution reveals two symmetrical branches, close to exponential. It is easy to see that the estimate $\langle \lambda^2 \rangle_{t_c \to \infty} = \int_0^\infty P_{T,t_c}^{(F)}(l)l^2 dl$, coincides with the result of Eq. 24.

This analysis reveals that in the underdamped regime the diffusion happens through flights of average squared length $\langle \lambda^2 \rangle_{t_c \to \infty} \propto t_c^2 \Delta U$, occurring at a rate $\Gamma_{t_c \to \infty} \propto e^{-\Delta U/T}/t_c$. Remarkably, the flight length and the flight rate do not depend on $L$, and the proportionality constants only weakly depends on the shape of the potential and on temperature.

### 3.6.2 Underdamped internal flights

The diffusion coefficient of the internal flights is given, in the underdamped limit, by $D_{t_c \to \infty}^{\cup} = \frac{1}{2t_c} P_{\text{d},t_c \to \infty} P_{\cap,t_c \to \infty} \langle (\Delta x^\cup)^2 \rangle_{t_c \to \infty}$. We have already determined $P_{\text{d},t_c \to \infty}$ and $P_{\cap,t_c \to \infty}$. Specifically, $P_{\text{d},t_c \to \infty} = 1$ as all the barrier crossing are uncorrelated, while $P_{\cap,t_c \to \infty}$ is given by Eq. 22. Thus, to determine the diffusion coefficient we need to calculate $\langle (\Delta x^\cup)^2 \rangle_{t_c \to \infty}$. In the underdamped regime, a particle performing a barrier crossing flight traverses many wells while traveling with an effective velocity $v(\epsilon)$, as discussed in the previous sessions, so that $\langle (\Delta x^\cap)^2 \rangle_{t_c \to \infty} \sim t_c^2$. Since, by definition, $\langle (\Delta x^\cup)^2 \rangle_{t_c \to \infty} \leq L^2$, we have $\langle (\Delta x^\cap)^2 \rangle_{t_c \to \infty} \gg \langle (\Delta x^\cup)^2 \rangle_{t_c \to \infty}$, so that in the underdamped regime the contribution of the internal flights to the diffusion coefficient is negligible, and $D_{t_c \to \infty} = D_{t_c \to \infty}^{\cap} + D_{t_c \to \infty}^{\cup} \simeq D_{t_c \to \infty}^{\cap}$. In Appendix A we show the result of the numerical evaluation of $\langle (\Delta x^\cup)^2 \rangle(t_c)$ on the whole $t_c$ range, determine a lower bound of this quantity for $t_c \to \infty$ and discuss the thermalization process of a particle after an external flight.

## 4 The overdamped to underdamped crossover

Traditionally in the BM, the overdamped and underdamped case are approached with different equations, being the slow and diffusing variable the position in the first case and the energy in the second [39]. More recently it has been shown that the two limits can be accessed through a singular perturbation expansion [40], performed using as a small parameter $\gamma$ or $1/\gamma$ respectively in the underdamped and overdamped limit. In both cases, the investigation of the features of the crossover between these two regimes is not accessible. Conversely, in the IK model, we have used a theoretical framework where the two component stochastic processes are by construction on the same footing, providing us with the convenient tool to investigate the crossover between the two dynamical regimes, which is what we do in the present section.

In particular, we will show that the transition between these two regimes is accompanied by a well-defined mathematical singularity that acts as a clear watershed. Specifically, the singularity characterizes the equilibrium probability measure, Eq. 10, of the flights, that start from a position $x_s$, acquire a total energy $E = \Delta U$, and end on the top of the barrier, $x_e = L/2$. We illustrate in Fig. 4 one of these critical flights in the phase portrait of the deterministic component of the dynamics, a Newtonian dynamics in the potential Eq. 1 not interrupted by collisions with the heat bath. The figure clarifies that the flights characterized by the singularity are are those at the edge between internal and external flights.

Without loss of generality, we describe this singularity focusing on the motion of a tracer with mass $m = 1$ in a well with $L = 2$ and $\Delta U = 1/4$. We consider the average squared length of the flights that reach the top of an energy barrier, whose probability distribution is given by

$$g(x_s, v_s) = f(x_s, v_s, t(x_s, v_s))(1 - x_s)^2. \tag{26}$$

This probability distribution allows identifying the flights mostly contributing to the diffusion

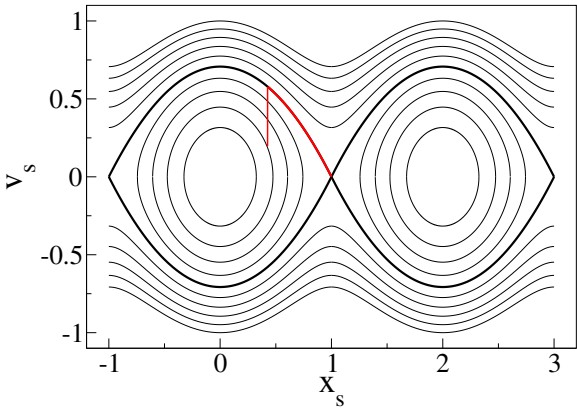

Figure 4: Phase portrait for a particle in the potential of Eq. 1, with $\Delta U = 1/4$ and $L = 2$. Lines correspond to $E/\Delta U = 0.2, 0.4, \ldots, 2$. The black bold line indicates the separatrix, $E/\Delta U = 1$. The function $f$ has a singularity corresponding to the flights on the separatrix that terminate on the top of the potential barrier, $x_e = 1$. The figure illustrates the evolution of a particle of energy $E/\Delta U = 0.4$ undergoing a collision in $x_s = 0.431$ following which it jumps on the separatrix and reaches $x_e = 1$ in a time $\Delta t \rightarrow \infty$.

coefficient as a function of $t_c$, i.e. as the system moves from the overdamped to the underdamped regime. Note that the duration $t(x_s, v_s)$ of the flights we are considering equals the time a particle in position $x_s$ with velocity $v_s$ needs to reach the top of the barrier. Fig. 5 shows the contour plot in of $g(x_s, v_s)$ for increasing value of $t_c$. First notice that the contour plots are bounded by the parabola

$$v_{s,min}(x_s) = \sqrt{\frac{2}{m}(\Delta U - V(x_s))}, \tag{27}$$

that equals the velocity of a particle in position $x_s$ with total energy $E = \frac{1}{2}mv_{s,min}^2(x_s) + V(x_s) = \Delta U$. This parabola is the separatrix represented as a thick line in the phase space portrait, Fig. 4.

Fig. 5 illustrates contour plot of $g$ for $t_c\omega_b = 0.15, 0.28, 0.8, 1.2$, from $a$ to $d$. Thus, in moving from panel $a$ to panel $d$ we are moving from the overdamped to the underdamped regime. In the overdamped regime, $g$ has a narrow maximum due to flights starting in the proximity of the top of the barrier. The maximum identifies a typical length scale and a typical energy scale for the flights contributing the most to the diffusion. This is consistent with the results of Sec. 3.5, where we have shown that these characteristic length and energy scales are those of the free flights. In this regime, the width of the maximum along the $x_s$ direction, as well as its distance from the top of the barrier, decreases like $t_c$, while the width of the maximum along the $v_s$ direction does not depend on $t_c$ but only on the temperature. As $t_c$ increases, we observe the maximum to widen, which means that flights starting from different positions $x_s$ inside the well contribute similarly to the mean square displacement. Moving towards the underdamped limit, the maximum spreads over a range of starting positions comparable to the well amplitude. These results signify that away from the overdamped limit the dynamics stop being characterized by flights with a typical length and with a typical energy scale. Formally, the transition from the overdamped limit can be investigated considering the limit of the function $g(x_s, v_s)$ for $\epsilon \rightarrow 0^+$, i.e. approaching the $v_{s,min}$ parabola from above. One finds this limit to be zero for $\omega_b t_c < 1$, and to diverge for $\omega_b t_c > 1$. In other words $g$ evaluated on the parabola $v_{s,min}$ vanishes for $\omega_b t_c < 1$, and diverges if $\omega_b t_c > 1$.

Fig. 6 illustrates the $\omega_b t_c$ dependence of the duration, the starting point, and the starting velocity of the flight maximizing $g$. In the overdamped regime, these quantities correspond

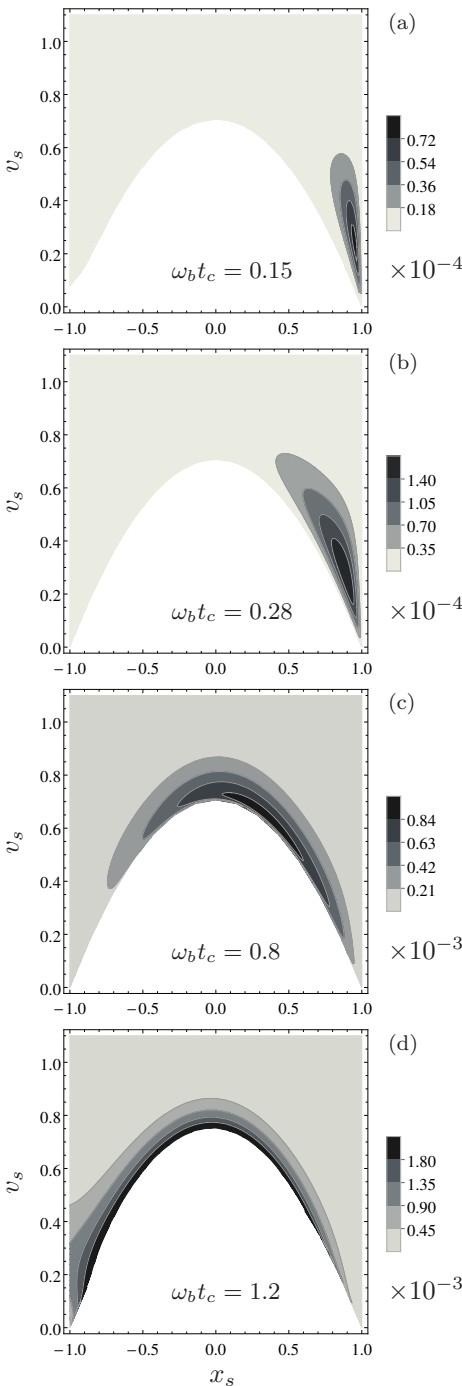

Figure 5: Contour plot of $g_{x_e=1}(x_s, v_s)$ for increasing values of $\omega_b t_c$. In the overdamped regime, $\omega_b t_c < 1$, $g_{x_e=1}(x_s, v_s)$ has a well defined maximum and vanishes on approaching the $E = \Delta U$ parabola. The maximum becomes broader as $t_c$ grows, as flights from a larger range of starting positions give a relevant contribution to $g$. At the transition between the overdamped and the underdamped regime, $\omega_b t_c = 1$, the maximum disappears as $g$ diverges logarithmically on approaching the $E = \Delta U$ parabola. This behavior persists in the underdamped regime. The disappearance of the maximum of $g$ indicates that the system loses a typical energy scale and a typical length scale on moving from the overdamped to the underdamped regime. The figure refers to $\Delta U = 1/4$, $L = 2$, $T/\Delta U = 0.21$ and $x_e = 1$.

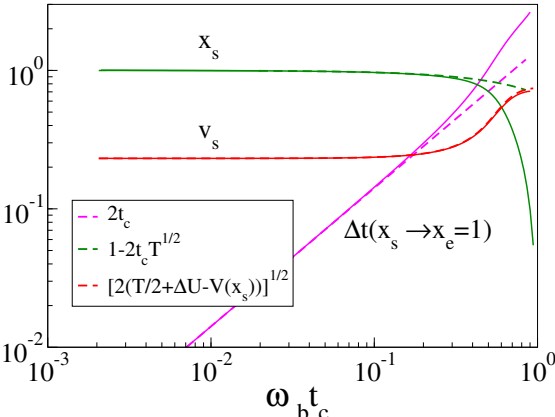

Figure 6: Starting position, starting velocity and time of flight of the maximum of $g$, illustrated in Fig.5, as a function of $\omega_b t_c$. These quantities (solid lines) depart from the free flights overdamped picture prediction (dashed lines) on approaching the crossover from the overdamped to the underdamped regime, $\omega_b t_c = 1$, where the maximum of $g$ is replaced by a divergence. The energy scale of the overdamped flights is $T/2$.

to those of a free flight (dashed lines) as expected: the flight duration scales as $2t_c$, and the velocity and the starting position are consistent with the particle kinetic energy being $T/2$. At $\omega_b t_c = 0.28$, corresponding to Fig. 5b, the free flight description starts to break down, and at $\omega_b t_c = 1$ the maximum disappears, being replaced by a divergence. As a summary, the graphical investigation of Figs. 5 and 6 suggests that the transition between the overdamped to underdamped regime happens through the loss of the overdamped energy and length scales, which leads to a discontinuity in $\omega_b t_c = 1$ of the function $g(x_s, v_s)$ when evaluated on the flights along the parabola $v_{s,min}$.

We now show that this interpretation is related to a logarithmic divergence of the equilibrium measure $f$, Eq. 10, on the critical flights. Specifically, we focus on flights on the separatrix that end in a generic position $x_e < 1$, and then consider the limit $x_e \to 1^-$ to investigate the flights that arrive on the top of the barrier. For these flights, the ratio $e^{-t_E(x_s \to x_e)/t_c}/v(x_e, E)$ appearing in the equilibrium measure becomes a ratio between vanishing quantities, as the time $t_E$ needed by a particle to reach the top of the barrier diverges, and the particle velocity on top of the barrier vanishes being $E = \Delta U$. While the investigation of the singular behavior can be carried out considering the actual potential, since the flights we are considering are heading to the top of the barrier and spending a large time in its proximity, it is convenient to approximate the potential close to its maximum with a parabola: $V(x) \simeq \Delta U - \frac{m\omega_b^2}{2}(1-x)^2$. Therefore the following discussion is essentially independent of the precise form of the potential, and only depends on $\omega_b$. In the parabolic approximation, we evaluate

$$\frac{e^{-t_E(x_s \to x_e)/t_c}}{v(x_e, E)} = \frac{1}{\omega_b} + \frac{1}{(1-x_s)^{\frac{1}{\omega_b t_c}}} + (1-x_e)^{\frac{1}{\omega_b t_c}-1}, \tag{28}$$

which in the $x_e \to 1^-$ limit equals

$$\lim_{x_e \to 1^-} \frac{e^{-t_E(x_s \to x_e)/t_c}}{v_{x_e}} = \begin{cases} 0, \omega_b t_c < 1 \\ t_c/(1-x_s), \omega_b t_c = 1 \\ \infty, \omega_b t_c > 1 \end{cases} \tag{29}$$

To summarize, in the overdamped regime, $t_c \omega_b < 1$, the probability that a flight reaches the top of a barrier is dominated by the flights that start at a distance $l = 2t_c\sqrt{T/m}$ from

the top, that have an energy $\Delta U + T/2$ and that reach the top in a time $2t_c$. Conversely, in the underdamped regime, $t_c \omega_b > 1$, flights starting from everywhere inside the well have a comparable probability to reach the top of the barrier, and this probability diverges in the $\epsilon \to 0^+$ limit. It is also possible to interpret $l$ as the spatial resolution around the top of the potential barrier needed in the overdamped regime to discern between an Il'in Kashminskii trajectory, that appears smooth at shorter lengthscale, and a BM trajectory, that is self-similar. For $t_c$ above the singularity, the smoothness of the trajectory is always already evident on a finite lengthscale of the order of the period of the potential.

It is worth remarking that, while $1/\omega_b$ is also the timescale separating the overdamped and the underdamped regime in the BM, the singular behavior described here has no analogous in the BM.

## 5 Brownian non-Gaussian dynamics

We now show that both the BM and the IK model in a periodic potential give rise to a Brownian non-Gaussian dynamics. That is, in both models, there is a long transient during which the displacement distribution, we will refer to as the van Hove distribution, is not Gaussian, while its variance, the mean square displacement, grows linearly in time. Our goal is to understand what are the timescales governing the relaxation of the dynamics towards its asymptotic scaling form. To investigate how long does it take for the Van Hove to acquire its asymptotic Gaussian shape, we investigate the time evolution of the excess kurtosis of the Van Hove distribution with respect to that of a Gaussian, for both the IK and BM processes.

The Brownian non-Gaussian dynamics is particularly relevant in the underdamped regime of both dynamics due to the coexistence of particles that have not yet left their original well, and of particles that have performed long barrier crossing flights. In this regime, the MSD$(\Delta t)$ can be considered as the average of the squared displacement until time $\Delta t$ over two random variables distributions. One is the number of external flights $n(\Delta t)$, until time $\Delta t$, and the other is the total displacement due to $n(\Delta t)$ barrier crossing flights. If the flights can be considered independent, the variance of the displacement after $n(\Delta t)$ flights is proportional to $n(\Delta t)$, whose probability distribution is $P_n(\Delta t) = \frac{e^{-\Delta t/t_c}}{\Gamma(n+1)} \left(\frac{\Delta t}{t_c}\right)^n$ and whose average, in turn, is proportional to $\Delta t$. Therefore the process will be Fickian (MSD$\propto \Delta t$). Instead, the process could be non Gaussian on an intermediate time, depending on the excess kurtosis of the single barrier crossing length distribution and on the number of flights, $n(\Delta t)$.

### 5.1 Overview

Figure 7 is an overview of the main features of the dynamics of the IK and BM models, in the overdamped regime, near the crossover and in the underdamped regime. In the next sections, we will deal in greater detail with the underdamped regime. The data have been obtained numerically averaging the dynamics over $2 \cdot 10^4$ to $2 \cdot 10^5$ particles, whose initial position is the equilibrium one.

The first column illustrates the evolution of the MSD, of the Van Hove distribution and of the ngp in the overdamped regime, with $\omega_0 \tau = 0.05$, where $\tau = t_c$ for the IK dynamics, and $\tau = \gamma^{-1}$ for the BM. In panels a, b, d and e the BM curve is not shown since it is essentially overlapped to the IK one. In the overdamped regime, the two dynamics have an indistinguishable MSD, as in Fig. 7a. At time $t \sim \tau$, the interaction with the heat bath affects the particle motion, that becomes diffusive within the well. At later times one observes a plateau due to the potential induced slow down of the dynamics, after which the dynamics enters its asymptotic diffusive regime, with a diffusion constant given by Eq. 16. Panel d illustrates the

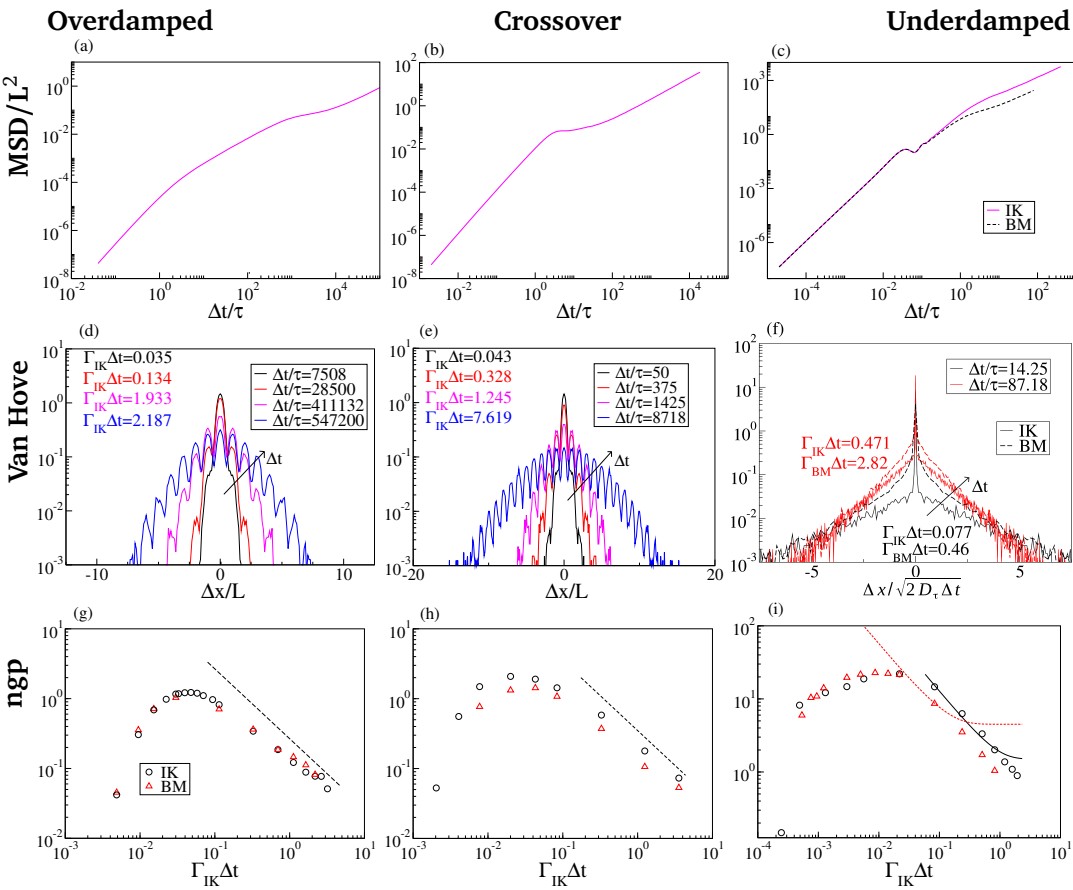

Figure 7: MSD, Van Hove distribution and ngp in overdamped regime ($\omega_b \tau = 0.05$), at the crossover ($\omega_b \tau = 1$) and in the underdamped regime ($\omega_b \tau = 100$), where $\tau = t_c$ in the IK model and $\tau = \gamma^{-1}$ in the BM. In panels a, b, d and e the BM curve is not shown since it is essentially overlapped to the IK one. $\Delta t/\tau$ is the number of collisions until time $\Delta t$ and $\Gamma_\tau \Delta t$ is roughly the number of uncorrelated external flights until time $\Delta t$. For $\omega_b \tau < 1$, $\Gamma_{\mathrm{IK}} \simeq \Gamma_{\mathrm{BM}}$. In panel f, we have coarse grained the Van Hove distribution of the two models over the wells, and we have normalized them using their diffusion constants. In panels g and h, dashed lines are guides for the eye of slope -1. In panel i, the solid line represents Eq. 38, and the dashed line Eq. 42. Parameters $L = 2$, $\Delta U = 1/4$ and $T/\Delta U = 0.21$.

Van Hove distribution of the two dynamics, which are also essentially identical, at different times, as in the legend. At short times, the Van Hove distribution is a Gaussian function reflecting the imposed initial Gaussian distribution of the velocities. When particle migrates to adjacent wells, the Van Hove distribution develops two wings. The timescale of this process is the inverse rate of uncorrelated barrier crossing flights, $\Gamma_\tau$. Thus, the quantity controlling the evolution of the distribution is the number of uncorrelated barrier crossing flights per particle, that is $\Gamma_\tau \Delta t$, we annotate close to the corresponding Van Hove. On a longer time scale, the distribution converges to a Gaussian. Consistently, the ngp illustrated in Fig. 7g grows at short times, and then decays, in a way numerically compatible with a power law with an exponent around $-0.83$ for both the IK and the BM.

The middle column of Fig. 7 illustrates the MSD, the Van Hove distribution and the ngp, in the crossover region, with $\omega_0 \tau = 1$. The MSD displays a ballistic regime, followed by a plateau due to the confining potential. Unlike in both the over and under damped regime the time of the beginning of the plateau coincides with the end of the ballistic regime due to the interaction with the heat bath. At longer times, the system reaches the asymptotic diffusive regime. The

Van Hove distribution of panel e is seen to evolve from a Gaussian to a Gaussian distribution, modulated by the periodicity of the potential, as time advances. Correspondingly, the ngp first rises from zero, and then decay to zero. The long-time decay of the ngp is compatible with a power law with an exponent around $-0.87$ for both the IK and the BM. As is in the overdamped regime, also in the crossover regime the IK and BM dynamics are extremely similar, but the ngp of the IK model is slightly larger than that of the BM.

In the underdamped limit, Fig. 7c, both particles not having enough energy to overcome the barrier and oscillating within the well, and particles that overcome the barrier, contribute to the MSD. At a time shorter than the oscillation period all particles contribute ballistically to the MSD, that scales as $T t^2$. At a longer time, the MSD becomes dominated by the particles that have enough energy to overcome the barrier and therefore scales as $T t^2 f$, where $f < 1$ is the fraction of particles with total energy $E > \Delta U$. Finally, on the time scale of interaction with the heat bath, the MSD enters the diffusive regime. In this regime, the IK and the BM models have a different diffusivity, given respectively by Eq. 21, and in Ref. [40]. Since in the underdamped regime particles generally travel long distances when escaping a well, it is convenient to coarse-grain over the wells the Van Hove, and to rescale it by its variance. Precisely, in panel f we rescale the x-axis by $\sqrt{2D_\tau \Delta t}$, where $D_\tau$ is the diffusion coefficient for the considered value of $\tau$, as at the considered values of $\Delta t$, $\Gamma_\tau \Delta t \gtrsim 10^{-1}$, the MSD has already reached its asymptotic diffusive behavior as shown in panel c, inset. For $\Gamma_\tau \Delta t < 1$ the Van Hove distribution has a high peak in zero surrounded by two exponential tails, as a consequence of the coexistence of many particles that are still in their original well, and of few particles that have performed barrier crossing flights. As time advances, the amplitude of the peak decreases, while that of the exponential tails increases. Correspondingly, the ngp also decreases, in a way compatible with a power law of exponent $-1$ for both the IK and the BM, as in panel i. However, in the underdamped regime, the convergence of the Van Hove towards a Gaussian occurs on a timescale which is not accessible in the simulations.

## 5.2 Short-Intermediate time ngp

We now introduce an analytical approach to describe the time dependence of the non Gaussian parameter in the underdamped limit, first for the IK model, and then for the BM, starting from a model for the time evolution of the Van Hove. For both processes, the Van Hove distribution is a Gaussian at both short and long times, while it acquires a different shape at intermediate times. Here we consider a model valid up to this intermediate timescale, while in Sec. 5.3 we will consider the long time behavior.

### 5.2.1 Van Hove of the IK model

At short times the Van Hove distribution $P(\Delta x, \Delta t)$ has contributions from the particles that have not escaped their original well, from particles that have performed one or more barrier crossing flights, and from the particles that are performing a barrier crossing flights. Here we assume that, after performing a barrier crossing flights, a particle thermalizes in the arrival well, which tantamount to neglecting the contribution to the Van Hove distribution of the particles that have performed more than a barrier crossing flight. This approximation is only approximately correct, as discussed in Appendix A, and we will therefore also suggest how this approximation could be relaxed. In this approximation, after a coarse-grain over the wells, the contribution of the particles that have not escaped appears as a $\delta$. The contribution to the Van Hove distribution from the particles that have performed and that are performing a barrier crossing flights at time $\Delta t$, is obtained from the displacement distribution of the particles that leave the initial well with an excess energy $\epsilon$ at time $r$, and collide again at time $s$, integrating over $\epsilon$. For $\Delta x > 0$ (the distribution is symmetric) the Van Hove distribution can be written

as:

$$P_\epsilon(\Delta x, \Delta t) = \delta(\Delta x)e^{-\Gamma_{\text{IK}}\Delta t} +$$
$$\int_0^{\Delta t} dr \Gamma_{\text{IK}} e^{-\Gamma_{\text{IK}}r} \left( \int_r^{\Delta t} ds \frac{e^{-\frac{s-r}{t_c}}}{t_c} \delta(\Delta x - v_{eff}(s-r)) + \int_{\Delta t}^\infty ds \frac{e^{-\frac{s-r}{t_c}}}{t_c} \delta(\Delta x - v_{eff}(\Delta t - r)) \right)$$
$$(30)$$

where $v_{eff}(\epsilon) = L/t_b(\epsilon)$ and $t_b(\epsilon)$ is defined in Eq. 18. By expliciting the two $\delta$ constraints we find

$$P_\epsilon(\Delta x, \Delta t) = \delta(\Delta x)e^{-\Gamma_{\text{IK}}\Delta t} + \frac{e^{-\frac{\Delta x}{v_{eff}t_c}}}{v_{eff}t_c} \left( 1 - e^{-\Gamma_{\text{IK}}\left(\Delta t - \frac{\Delta x}{v_{eff}}\right)} \theta\left(\Delta t - \frac{\Delta x}{v_{eff}}\right) \right) +$$
$$e^{-\Gamma_{\text{IK}}\Delta t} \frac{\Gamma_{\text{IK}}t_c}{1 - \Gamma_{\text{IK}}t_c} \frac{e^{-\frac{\Delta x(1-\Gamma_{\text{IK}}t_c)}{v_{eff}t_c}}}{v_{eff}t_c}(1 - \Gamma_{\text{IK}}t_c), \quad (31)$$

where the first contribution represents the particles not yet escaped, the second represents the escaped particles and the third represents the escaping particles. Eq. 31 can be rearranged as

$$P_\epsilon(\Delta x, \Delta t) = \delta(\Delta x)e^{-\Gamma_{\text{IK}}\Delta t} + \left[ \frac{e^{-\frac{\Delta x}{v_{eff}t_c}}}{v_{eff}t_c} - e^{-\Gamma_{\text{IK}}\Delta t} \frac{e^{\frac{-\Delta x(1-\Gamma_{\text{IK}}t_c)}{v_{eff}t_c}}}{v_{eff}t_c}(1 - \Gamma_{\text{IK}}t_c) \right] \theta\left(\Delta t - \frac{\Delta x}{v_{eff}}\right).$$
$$(32)$$

Since we work at low temperature, we can neglect $\Gamma_{\text{IK}}t_c \sim e^{-\frac{\Delta U}{T}}$ with respect to 1. Physically, this also implies that we are neglecting the amount of escaping particles with respect to the amount of those that have been already escaped, as the average time of flight $t_c$ is much shorter than the timescale of residence in a well: $\Gamma_{IK}^{-1} \sim t_c e^{\frac{\Delta U}{T}} \gg t_c$. Considering both positive and negative $\Delta x$ and averaging over the excess energy distribution $P(\epsilon) = e^{-\epsilon/T}t_b(\epsilon)/\int_0^\infty e^{-\epsilon/T}t_b(\epsilon)d\epsilon$, yields

$$P(\Delta x, \Delta t) = e^{-2\Gamma_{\text{IK}}\Delta t}\delta(\Delta x) + (1 - e^{-2\Gamma_{\text{IK}}\Delta t})P^{(F)}(\Delta x, \Delta t) \quad (33)$$

where

$$P^{(F)}(\Delta x, \Delta t) = N(\Delta t) \left\langle \frac{e^{-\frac{|\Delta x|}{v_{eff}t_c}}}{2v_{eff}t_c} \theta\left(\Delta t - \frac{|\Delta x|}{v_{eff}}\right) \right\rangle_\epsilon$$
$$= N(\Delta t) \int_0^\infty e^{-\frac{\epsilon}{T} - \frac{|\Delta x| \, t_b(\epsilon)}{Lt_c}} t_b^2(\epsilon) \theta\left(\Delta t - \frac{|\Delta x| \, t_b(\epsilon)}{L}\right) d\epsilon. \quad (34)$$

is the distribution of the distance traveled by the particles that have left their original well in a time $\Delta t$, and $N$ is a normalization factor.

The distribution of the flight lengths $P^{(F)}(\Delta x)$, Eq. 25, is thus recovered from Eq. 34 by neglecting the $\theta$ function. Physically, this is the approximation of instantaneous flights, that leads to a continuous time random walk (CTRW) description of the dynamics. This approximation is reasonable, as for instance the single flight distribution, Eq. 25, decays roughly exponentially at large $\Delta x$, alike the Van Hove distribution of Fig. 7f. In this approximation, the Van Hove distribution is described as the sum of a delta function in the origin and of an exponential like distribution $P^{(F)}(\Delta x)$,

$$P_{\text{IK}}(\Delta x, \Delta t) = e^{-2\Gamma_{\text{IK}}\Delta t}\delta(\Delta x) + (1 - e^{-2\Gamma_{\text{IK}}\Delta t})P^{(F)}(\Delta x), \quad (35)$$

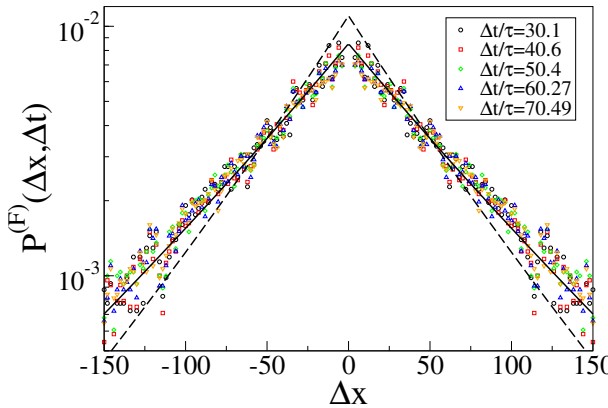

Figure 8: For the IK model, we show numerical results for the normalized probability distribution $P^{(F)}(\Delta x, \Delta t)$ at different times, as obtained from the Van Hove distribution by removing the $\delta$-peak in the origin. The dashed line represents the theoretical prediction for $P^{(F)}(\Delta x, \Delta t)$ of Eq. 34, while the full line is the rescaled distribution $(1/\alpha)P^{(F)}(\alpha\Delta x, \Delta t)$ with $\alpha = 1.3$.

or, for $\Gamma_{\text{IK}}\Delta t \ll 1$,

$$P_{\text{IK}}(\Delta x, \Delta t) = (1 - 2\Gamma_{\text{IK}}\Delta t)\delta(\Delta x) + 2\Gamma_{\text{IK}}\Delta t P^{(F)}(\Delta x). \tag{36}$$

In Sec. 5.3 we will show that this functional form for the Van Hove distribution is derived at short times within a CTRW formalism where an exponential distribution of the length of the single flight, $P^{(F)}(\Delta x)$, is assumed. Note that, in this instantaneous flight approximation, the only time dependence of the Van Hove distribution is in the relative weight of its two terms.

When the instantaneous flight approximation is relaxed, the Van Hove distribution has an additional time dependence due to that of $P^{(F)}(\Delta x, \Delta t)$. In particular, the actual distribution of the distance of the flying particles differ from the distribution of the flight length away from the origin, for $|\Delta x| \gtrsim \frac{L\Delta t}{t_b(\langle\epsilon\rangle)}$ where $\langle\epsilon\rangle \sim T$. For this values of $\Delta x$, $P^{(F)}(\Delta x, \Delta t)$ has a faster than exponential decay, while $P^{(F)}(\Delta x)$ has a roughly exponential decay. This has some consequences. In particular, with respect to the ngp of $P^{(F)}(\Delta x)$, that of $P^{(F)}(\Delta x, \Delta t)$ is dominated by smaller values of $\Delta x$, and its estimation is, therefore, less affected by the finite statistics. In addition, while the ngp of $P^{(F)}(\Delta x)$ is constant, that of $P^{(F)}(\Delta x, \Delta t)$ slowly grows in time, approaching that of $P^{(F)}(\Delta x)$.

Fig. 8 shows that the predicted displacement distribution, Eq. 34 (dashed line), does not compare well with the measured one, which is obtained subtracting from the Van Hove the delta peak in zero. This has to be expected, as we have derived Eq. 32 assuming the particles that perform a flight to thermalize in the arrival well, while in Appendix A we will show that at least 30% of them do not thermalize, but rather performs two or more barrier crossing flights in sequence. While it has already been shown that this circumstance does not significantly affect the diffusion constant [42], Fig. 8 reveals that it does affect the shape of the displacement distribution at short times. If a particle performs more than a flight in a time $\Delta t$, then the particles will perform flights of short duration, and thus with a small length $\Delta x$. This implies that the estimated Van Hove overestimates the number of particles with small displacements, and underestimate that with large displacements. Numerically, we observe that the actual Van Hove can be simply described through a rescaling of the distribution by a factor $\alpha = 1.3$, as the function $(1/\alpha)P^{(F)}(\alpha\Delta x, \Delta t)$ (full line) correctly describes the data, as illustrated in Fig. 8.

The above rescaling of Eq. 34 does not affect the non Gaussian parameter

$$\text{ngp}(\Delta t) = \frac{\langle\Delta x^4\rangle}{3\langle\Delta x^2\rangle^2} - 1, \tag{37}$$

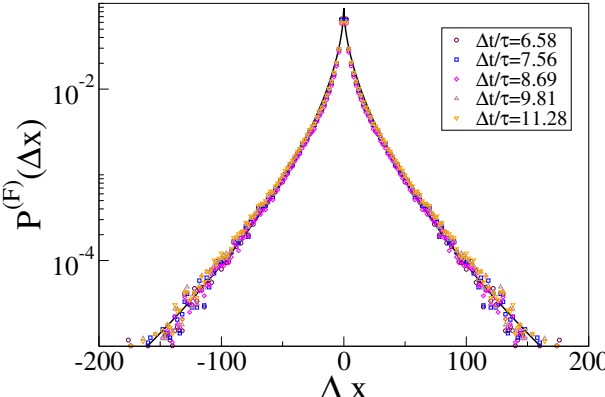

Figure 9: Normalized probability distributions $P^{(F)}(\Delta x)$, as obtained normalizing the Van Hove after removing the $\delta$ in the origin, at different times. The excellent collapse of the distributions confirms that in the considered temporal range the Van Hove distribution is dominated by the escaped particles. The two parameters fit Eq. 40 is in very good agreement with data.

which results

$$\mathrm{ngp}(\Delta t) = \frac{k_{\mathrm{IK}}(\Delta t) + 1}{1 - e^{-2\Gamma_{\mathrm{IK}}\Delta t}} - 1. \tag{38}$$

Neglecting the small dependence on $\Delta t$, and considering that $k_{\mathrm{IK}} \simeq 1.5$, the parameter is found to decrease in time as in Fig. 7i (solid line). At short times, $\Gamma_{\mathrm{IK}}\Delta t \ll 1$ we find

$$\mathrm{ngp}(\Delta t) = \frac{k_{\mathrm{IK}} + 1}{2\Gamma_{\mathrm{IK}}\Delta t}, \tag{39}$$

which shows that the ngp decays as $\Delta t^{-1}$.

### 5.2.2 Van Hove distribution of the BM model

In the BM the ratio between the timescale of a barrier crossing event, $\gamma^{-1}$, and the average residence time, $\Gamma_{\mathrm{BM}}^{-1} = \frac{T}{\Delta U \gamma} e^{\frac{\Delta U}{T}}$, is $\frac{\Delta U}{T} e^{-\frac{\Delta U}{T}}$. This ratio, although larger than the corresponding one for the IK, $e^{-\frac{\Delta U}{T}}$, still vanishes in the low temperature limit. This suggests that while at very short times the tails of the Van Hove distribution are dominated by the escaping particles (not shown), later on, when the ngp decays as $\Delta t^{-1}$ as in Fig. 7i, the contribution of escaping particles to the Van Hove distribution could be negligible. If this is so, in this time regime also in the BM the Van Hove is the sum of the displacement distribution of the trapped particles, that at a coarse-grained level is represented by $\delta(\Delta x)$, and of the displacement distribution of the particles that have performed a flight, whose relative weights change as time advances. To check this assumption, we show in Fig. 9 that the Van Hove at different times, deprived of the $\delta$ peak and normalized, which should equal the time independent single flight distribution, do indeed collapse on a master curve. This curve, not known analytically due to the absence of a solution of the associated FPE, turns out to be well described by a stretched-exponential functional form,

$$P_{\mathrm{BM}}^{(F)}(\Delta x) = \frac{e^{-A|\Delta x|^B}}{2A^{-1/B}\Gamma(1 + \frac{1}{B})}, \tag{40}$$

with best fit parameters $A = 0.473 \pm 0.02$ and $B = 0.582 \pm 0.008$, as illustrated in Fig. 9. Thus, in the considered instantaneous flight approximation, whose validity is limited as for the IK model, the time evolution of the Van Hove distribution is

$$P_{\mathrm{BM}}(\Delta x, \Delta t) = e^{-2\Gamma_{\mathrm{BM}}\Delta t}\delta(\Delta x) + (1 - e^{-2\Gamma_{\mathrm{BM}}\Delta t})P_{\mathrm{BM}}^{(F)}(\Delta x). \tag{41}$$

In analogy with Eq. 38 we find:

$$\text{ngp}(\Delta t) = \frac{k_{\text{BM}}(\Delta t) + 1}{1 - e^{-2\Gamma_{\text{BM}}\Delta t}} - 1. \tag{42}$$

If we fix $k_{\text{BM}} = 4.595$, as numerically evaluated from Eq. 40 for the given values of the parameters $A$ and $B$, Eq. 42 is represented by the dashed line in Fig. 7i.

### 5.2.3 Short times Van Hove comparison

At short time, $\Delta t/\tau < 1$, the IK and the BM dynamics have the same mean square displacement, as in Fig. 7c. However, they do have a different Van Hove distribution. To compare the two dynamics, we consider the ratio between the ngp of the two models. If $k_{(\text{IK,BM})}$ is the ngp of $P^{(F)}_{(\text{IK,BM})}(\Delta x)$, Eqs. 38 and 42 at short times yield

$$\frac{\text{ngp}_{\text{IK}}}{\text{ngp}_{\text{BM}}} = \frac{k_{\text{IK}} + 1}{k_{\text{BM}} + 1}\frac{\Gamma_{\text{BM}}}{\Gamma_{\text{IK}}}.$$

The temperature dependence of this quantity is mainly determined by the ratio of the rates IK and BM, as given in Eq. 23 and in [39] respectively, since the temperature dependence of $k_{\text{IK}}$ and $k_{\text{BM}}$ at low temperature is quite weak. For instance, at $T/\Delta U = 0.21$ $\frac{\Gamma_{\text{BM}}(T)}{\Gamma_{\text{IK}}(T)} \simeq 6.36$, while $k_{\text{BM}} \simeq 4.6$ and $k_{\text{IK}} \simeq 1.51$, so that $\frac{\text{ngp}_{\text{IK}}}{\text{ngp}_{\text{BM}}} \simeq 2.85 > 1$. This implies that the IK follows the BM in the decrease of the ngp as from Fig. 7i.

## 5.3 Long times ngp and CTRW

At long times, $\Gamma\Delta t \gg 1$, particles have performed many flights, and the Van Hove distribution converges towards its asymptotic Gaussian shape. While the associated timescale is not accessible in simulations in the underdamped limit, yet it is possible to make analytical progress as concern the time evolution of the ngp. Here we show that this is the case first considering the IK model, where the time dependence of the ngp, in both the short and the long time limit, can be derived analytically within a continuous time random walk treatment, and then generalizing the result to the BM.

In the IK model we can assume the waiting time distribution, which is the distribution of the residence time in a well, to be $\psi(t_w) = \frac{1}{\tau_{esc}}e^{-\frac{t_w}{\tau_{esc}}}$, where $\tau_{esc} = 1/2\Gamma_{\text{IK}}$ is the timescale of escape from the well, regardless of the escaping direction. The single flight distribution, according to the considerations of the previous sections, is assumed to be $\phi(L) = \frac{1}{2L_c}e^{-\frac{|L|}{L_c}}$ [9]. The Montroll-Weiss equation for the Fourier transform in the space and Laplace in the time of the displacement distribution is:

$$P(k,s) = \frac{1 - \tilde{\psi}(s)}{s}\frac{1}{1 - \tilde{\psi}(s)\hat{\phi}(k)}, \tag{43}$$

where $\tilde{\psi}(s) = \frac{1}{\tau_{esc}s}$ is the Laplace transform of $\psi(t_w)$ and $\hat{\phi}(k) = \frac{1}{1+k^2L_c^2}$ is the Fourier transform of $\phi(L)$. The Laplace inversion of Eq. 43 gives:

$$\hat{P}(k,\Delta t) = e^{-\frac{\Delta t}{\tau_{esc}}\frac{k^2L_c^2}{1+k^2L_c^2}}. \tag{44}$$

The second and fourth moment are obtained by derivation:

$$\langle \Delta x^2(\Delta t) \rangle = -\partial_k^2\hat{P}(k,\Delta t)|_{k=0} = \frac{2L_c^2}{\tau_{esc}}\Delta t, \tag{45}$$

$$\langle \Delta x^4(\Delta t) \rangle = \partial_k^4\hat{P}(k,\Delta t)|_{k=0} = \frac{4L_c^4(3\Delta t + 6\tau_{esc})}{\tau_{esc}^2}\Delta t. \tag{46}$$

Thus, the non Gaussian parameter results

$$\text{ngp}_{\text{IK}} = \frac{\langle \Delta x^4(\Delta t) \rangle}{3 \langle \Delta x^2(\Delta t) \rangle^2} - 1 = \frac{2\tau_{esc}}{\Delta t} = \frac{\Delta t^{-1}}{\Gamma_{\text{IK}}}. \tag{47}$$

This is the same result of Eq. 39 when $k_{\text{IK}} = 1$, which is the ngp of the exponential. This is not surprising, as Eq. 39 can be derived directly from Eq. 36, that is the short time limit of Eq. 44. Indeed, expanding Eq. 44 to the lowest order for $\Delta t \ll \tau_{esc}$ and inverting back to the position space, one finds

$$P(\Delta x, \Delta t) \simeq \delta(\Delta x) \left( 1 - \frac{\Delta t}{\tau_{esc}} \right) + \frac{\Delta t}{\tau_{esc}} \frac{1}{2L_c} e^{-\frac{|\Delta x|}{L_c}}, \tag{48}$$

which is equivalent to Eq. 36 when $\Gamma_{\text{IK}} \Delta t \ll 1$ and for not too far tails. Moreover, the above derivation of Eq. 47 is valid at all times, and in particular for asymptotically large times.

More generally, for large times, irrespectively of the dynamics, the displacement distribution can be seen as the sum of independent identically distributed single flights. Therefore the only relevant factors are the shape of the single flight displacement distribution through its ngp and the rate of escape. By exploiting the additivity of the second and fourth cumulants, equally for the BM and the IK model one obtains:

$$\text{ngp} = k_{\text{IK,BM}} / \Gamma_{\text{IK,BM}} \Delta t, \tag{49}$$

where $k_{\text{IK,BM}}$ is the ngp of the single flight displacement. Therefore the ngp ratio results:

$$\frac{\text{ngp}_{\text{IK}}}{\text{ngp}_{\text{BM}}} = \frac{k_{\text{IK}}(\Delta x_1)}{k_{\text{BM}}(\Delta x_1)} \frac{\Gamma_{\text{BM}}}{\Gamma_{\text{IK}}}. \tag{50}$$

## 6 Discussion

The Brownian model describes the stochastic motion of a particle in the limit in which the interaction with the heat bath occurs continuously in time. In the presence of a confining potential, that introduces an additional timescale related to its curvature, $\omega_b^{-1}$, the associated Langevin and Fokker-Plank equations result difficult to solve for generic values of the damping parameter, $\gamma$. Indeed, the overdamped, $\gamma \gg \omega_b^{-1}$, and the underdamped $\gamma \ll \omega_b^{-1}$ solutions are obtained by performing different approximations [39, 40], which are difficult to link so as to gain insights into the physical processes occurring when the system moves from one regime to the other. To better understand the physics of diffusive processes at the crossover between the overdamped and the underdamped regimes, here we have investigated the Il'in Khasminskii model for the diffusion of a particle within a potential. This model is introduced considering that the motion of a particle or the evolution of another dynamical variable describing a physical system, that eventually diffuses, is never Brownian at all the timescales. Indeed, particles are expected to interact with the heat bath with a finite rate, e.g. the rate of collisions with surrounding molecules $1/t_c$, not continuously in time. In this respect, the model is quite similar to a random walk. We have developed a theoretical framework to investigate the dynamics of IK model for a particle confined in a periodic potential, assuming the bath particles and the tracer to have the same mass, which implies that $t_c$ also plays the role of the thermalization timescale, which in the Brownian model is the inverse viscosity $\gamma^{-1}$. We have shown that the dynamics is conveniently described as resulting from the superposition of two related stochastic processes, one describing the motion within the potential wells, the other the transition between different wells. These processes dominate the diffusion in the overdamped and in the underdamped regimes, respectively [42].

Our approach treats on the same footing the different damping regimes, at variance with the theoretical approaches developed to investigate the Brownian dynamics, and thus put us in the unique position of investigating physics processes associated to the crossover from the overdamped to the underdamped regime. In this respect, we have shown that in the overdamped regime the dynamics is characterized by a typical time scale and by a typical length scale, which are those of the free flights, that are lost in the underdamped regime. In particular, the crossover between these two regimes is associated to a singularity in the probability of observing a particle to perform a flight ending on top of the barrier separating two wells. The search of a similar behaviour in other contexts is of interest within the general theory of threshold phenomena, where the relevant physics is played on a separatrix. The case of neuronal activity [50] could be an example.

Our approach also allows investigating the time evolution in the overdamped regime, where the system exhibits a Brownian non-Gaussian dynamics whereby a diffusive mean square displacement coexists with a non-Gaussian displacement probability distribution, for a long transient. The Brownian non-Gaussian dynamics, which is shared by a variety of different systems, is currently explained through coarse-grained models assuming the coexistence of particles with different diffusivities [13,43,44], or time-dependent diffusivities [27,45,46]. Here we have shown that it is possible to rationalize this dynamics without making any assumption on the diffusivity of the particles. Specifically, we have clarified that the Brownian non-Gaussian emerges when the displacement probability distribution is affected by the two stochastic processes we have decomposed the dynamics into. Indeed, we find that at short times the displacement distribution is dominated by the stochastic processes describing the motion within the potential wells, while at longer times it is dominated by the stochastic process describing the transition between different wells. As a consequence, the associated non-Gaussian parameter displays a maximum and decays at long times with a time scale fixed by the escape rate of a particle from a well.

As a final remark, we would like to comment on the relation between the results we have found for the IK model, and those expected for the BM, for which it is more difficult to make analytical progress. To this end, we recall that the diffusion of a particle in a potential is characterized by three timescales, which are the timescale $\omega_b^{-1}$ set by the potential, the thermalization timescale $t_{\text{therma}}$, and the inverse rate of interaction with the heat bath $t_c$. These three timescales allow comparing the low-temperature dynamics of the two models as a function of two non-dimensional ratios, such as $t_c \omega_b$ and $t_{\text{therma}} \omega_b$. For the IK model, we have considered $t_{\text{therma}} = t_c$, so that we have effectively investigated the diagonal of Fig. 10. In general, since the thermalization is due to the collisions with the heat bath, $t_c < t_{\text{therma}}$. The BM dynamics is a limiting one for which $t_c = 0$. Thus, the BM describes the $x$-axis of the diagram illustrated in Fig. 10. However, it is reasonable to expect the BM results to hold as long as $t_c$ is much smaller than the thermalization timescale. We have actually found differences in the low-temperature time evolution of the displacement distribution of the IK model and of the BM (x-axis) only in the deep underdamped regime, and thus speculate that the two models exhibit an analogous behavior in the whole shaded area of Fig. 10. In the underdamped regime, the two models have a different dependence of the escape rate and of the diffusion constant on temperature, which implies that on decreasing the temperature their difference increases. However, we remark that these differences only increase in the underdamped regime, as the two dynamics are identical in the overdamped regime. In particular, we showed analytically in Eq. 29 that the crossover value of $t_c$ does not depend on temperature. Thus, the two dynamics match at the crossover between the overdamped and the underdamped regime for all temperatures. This has an important consequence. Indeed, while in the BM it is possible to define a thermal length in the overdamped regime, starting with the Smoluchowski equation, it is difficult to formally identify typical scales at the crossover. Nevertheless, our results suggest that in the

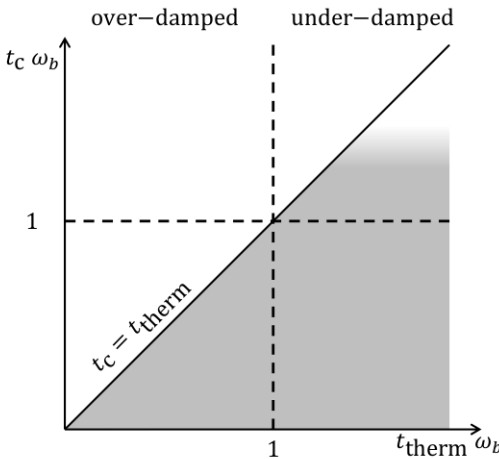

Figure 10: Schematic comparison between the BM and the IK models, in the $t_c\omega_b-t_{\text{therma}}\omega_b$ phase diagram, where $t_{\text{therma}}$ is the thermalization timescale, $t_c$ is the inverse rate of interaction with the heat bath, and $\omega_b^{-1}$ is the timescale set by the confining potential. In the Brownian model, the interaction with the heat bath occurs continuously in time, and $t_c = 0$. More generally, one expects the BM description of the dynamics to approximately hold as long as $t_c \ll t_{\text{therma}}$. Our numerical results show that at low temperature the particle displacement distributions of the BM and of the IK model with $t_c = t_{\text{therma}}$ only differ in the highly underdamped regime. We thus speculate that the two models behaves similarly in the shaded area of the diagram. Importantly, the displacement distribution function of the two models is indistinguishable in the crossover region at all times. Thus, as for the IK model, also for the BM we expect the crossover between the overdamped and the underdamped dynamics to be associated to the disappearance of a typical length scale and of a typical energy scale.

BM the crossover between the overdamped and the underdamped regime is associated with a disappearing length scale and to a disappearing time scale, as for the IK model.

In the future, it would be interesting to extend our formalism, we have developed for $t_c = t_{\text{therma}}$ as in Fig. 10, to the generic case $t_c < t_{\text{therma}}$. This would provide, by taking the limit $t_c \to 0$, a novel theoretical approach to investigate the features of the BM dynamics.

## Acknowledgments

Support from the Singapore Ministry of Education through the Academic Research Fund (Tier 1) under Projects No. RG104/15 and RG179/15 is gratefully acknowledged.

## A  Lower bound for $\langle(\Delta x^\cup)^2\rangle_{t_c\to\infty}$

In this Appendix we evaluate a lower bound for the mean square displacement of the internal flights in the underdamped limit, $\langle(\Delta x^\cup)^2\rangle_{t_c\to\infty}$, and discuss the thermalization process of a particle in the well where it undergoes a collision.

The average square distance between $x_e$ and $x_s$ is given by

$$\langle (\Delta x^\cup)^2 \rangle_{t_c \to \infty} = \int_{-L/2}^{L/2} dx_e dx_s P_e(x_e) P_s(x_s|x_e)(x_s - x_e)^2,$$

where $P_s(x_s|x_e)$ is the conditional probability that a particle enters a well landing in position $x_e$, and leaves the well through a flight starting from position $x_s$. Considering that the particle travels from $x_e$ to $x_s$ by forming a sequence of $k \geq 1$ flights, one has

$$\langle (\Delta x^\cup)^2 \rangle_{t_c \to \infty} = \int_{-L/2}^{L/2} dx_e dx_s (x_s - x_e)^2 P_e(x_e) \sum_{k=1}^{\infty} p_k P_s^{(k)}(x_s|x_e), \tag{51}$$

where $P_s^{(k)}(x_s|x_e)$ is the conditional probability that a particle entered in position $x_e$, leaves the well through a flight starting in position $x_s$, after interacting $k$ times with the heat bath.

Since in the underdamped limit barrier crossing flights are much longer than the potential period $L$, the probability distribution that a flight ends in an interval $x_e, x_e+dx_e$ is proportional to the time the particle needs to traverse that interval, $dx_e/v(x_e)$. Accordingly,

$$P_e(x_e) = \frac{1}{h(T)} \int_0^{\infty} \frac{e^{-\frac{\epsilon}{T}} t_b(\epsilon)}{\sqrt{\frac{2}{m}(\Delta U + \epsilon - V(x_e))}} d\frac{\epsilon}{m}, \tag{52}$$

where $h(T) = \int_0^{\infty} e^{-\frac{\epsilon}{T}} t_b^2(\epsilon) d\epsilon$. This probability distribution has two peaks, located at the extreme of the wells, as there the velocity of the particle is smaller.

The conditional probability $P_s^{(k)}(x_s|x_e)$ appearing in Eq. 51 is difficult to calculate. As $k$ increases the distribution evolves from $P_s^{(1)}(x_s|x_e) = \delta(x_s - x_e)$ to $P_s^{(th)}(x_s)$, which is the probability that a particle in thermal equilibrium performs a barrier crossing flight starting from position $x_s$,

$$P_s^{(th)}(x_s) \propto e^{-\frac{V(x_s)}{T}} \left[ 1 - \text{erf}\left( \sqrt{\frac{\Delta U - V(x_s)}{T}} \right) \right]. \tag{53}$$

After how many collisions $k$ we can assume $P_s^{(k)}(x_s|x_e) \simeq P_s^{(th)}(x_s)$? We numerically estimate the number of collisions needed for $P^{(k)}(x_s|x_e)$ to converge to $P_s^{(th)}(x_s)$ in Fig. 11, where the distribution of the total number of collisions in a well is shown. An initial spike over the exponential distribution of a thermalized particle is evident, indicating an enhanced probability of escape for $k \leq 5$. For $k = 1$ $\langle (\Delta x^\cup)^2 \rangle_{k=1} = 0$ and for $k \geq 6$ we can consider the particle thermalized. We now consider that the probability that a particle exists the well before thermalizing, and thus contributing to the square displacement an amount $\langle (\Delta x^\cup)^2 \rangle_k > \langle (\Delta x^\cup)^2 \rangle_\infty$, amounts to roughly 10%. Therefore, neglecting the correlations between $x_s$ and $x_e$ for all $k$, that is using $\langle (\Delta x^\cup)^2 \rangle_{k \to \infty}$ for all $k > 0$, gives us a lower bound for $\langle (\Delta x^\cup)^2 \rangle_{t_c \to \infty}$:

$$\sum_{k=2}^{\infty} \int_{-L/2}^{L/2} dx_e dx_s (x_s - x_e)^2 P_e(x_e) p_k P_s^{(k)}(x_s|x_e)$$

$$\simeq (1 - p_1) \int_{-L/2}^{L/2} dx_s dx_e P_e(x_e) p_s^{(th)}(x_s)(x_s - x_e)^2$$

$$= (1 - p_1) \langle \Lambda^2 \rangle_{t_c \to \infty}$$

$$= (1 - p_1)\left( L_e(T) + L^{(th)}(T) \right). \tag{54}$$

Here $L_e(T) = \int_{-L/2}^{L/2} P_e(x_e) x_e^2 dx_e$ is the squared width of the distribution of the position of the first interaction in the well and $L^{(th)}(T) = \int_{-L/2}^{L/2} p_s^{(th)}(x_s, T) x_s^2 dx_s$ is the squared width of

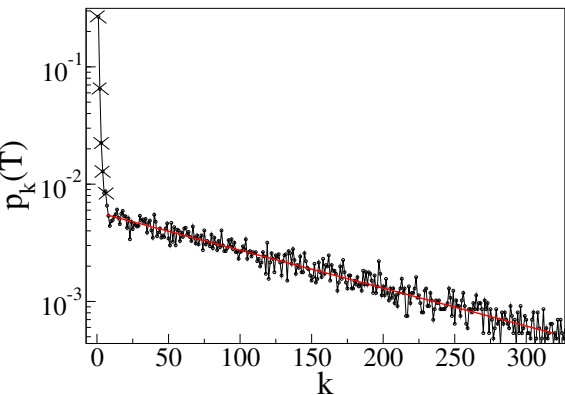

Figure 11: Distribution of the number of collisions a particle performs in a given potential well before escaping. The peak form for small $k$ values is associated to the particles that exit from a well before thermalizing into it. Thermalization is seen to occur after $k \simeq 6$ collisions. Parameters: $\Delta U = 1/4$, $L = 2$, $T/\Delta U = 0.21$ and $\omega_b t_c = 18$.

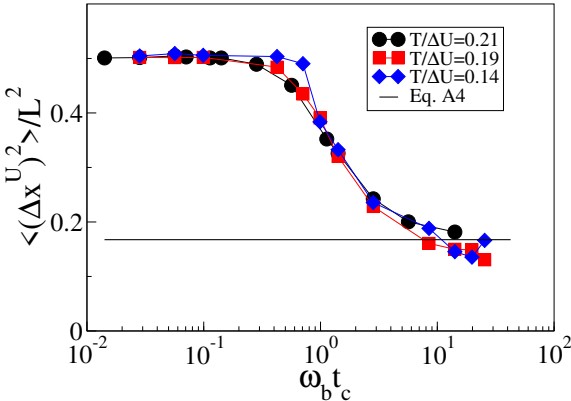

Figure 12: Dependence of the average squared length of the internal flights on $\omega_b t_c$. For particles that perform $k = 1$ collisions in the arrival well before escaping, $\Delta x^{\cup} = 0$. The full line is a lower bound determined under the assumption that a particle thermalizes after performing $k > 1$ collisions. The $\omega_b t_c \to 0$ limit is $L^2/2$. See Appendix A. Parameters: $\Delta U = 1/4$, $L = 2$.

the distribution of the position of the last interaction in the well before escaping. Both these terms are proportional to $L^2$, so that

$$\langle (\Delta x^{\cup})^2 \rangle_{t_c \to \infty} \simeq p_1 \cdot 0 + (1 - p_1(T)) \langle \Lambda^2 \rangle_{t_c \to \infty} \propto (1 - p_1) L^2. \tag{55}$$

Here $p_1$ is the probability that the particle exits from the well after a single interaction with the heat bath in position $x_s = x_e$. In the underdamped limit, this is the probability that after interacting with the heat bath the particles acquires an energy $E > \Delta U$,

$$p_1 = \left\langle \left( 1 - \text{erf}\left( \sqrt{\frac{\Delta U - V(x_e)}{T}} \right) \right) \right\rangle_{P_e} = \frac{1}{h(T)} \int_0^\infty d\frac{\epsilon}{m} \int_{-L/2}^{L/2} dx_e \frac{e^{-\frac{\epsilon}{T}} t_b(\epsilon)}{\sqrt{\frac{2}{m}(\Delta U + \epsilon - V(x_e))}} \times$$
$$\left( 1 - \text{erf}\left( \sqrt{\frac{\Delta U - V(x_e)}{T}} \right) \right). \tag{56}$$

At $T/\Delta U = 0.21$, we found $p_1 \simeq 0.3$. In principle it is easy to write analogously the expression of $p_k$ (or also of $P_s^{(k)}(x_s|x_e)$) for $k > 1$, that is, however, a $2k$-dim integral, so that for $k > 2$

the actual calculation becomes immediately prohibitive. Using the above result we find

$$D^{\cup}_{t_c \to \infty} = \frac{P_{\cap, t_c \to \infty}}{2 t_c} \langle (\Delta x^{\cup})^2 \rangle_{t_c \to \infty} \propto \frac{e^{-\frac{\Delta U}{T}}}{t_c} L^2. \tag{57}$$

The constant in front of Eq. 57 has a complicated expression we do not report here.

To check our approximation, we investigate in Fig. 12 the squared flight length of the internal flights as a function of $t_c \omega_b$. In the underdamped limit, we see $\langle (\Delta x^{\cup})^2 \rangle_{t_c \to \infty}$ to approach a constant, which is only weakly temperature dependent, that is compatible with the lower bound we have estimated (full line).

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
