# Peer review of "Unifying description of the damping regimes of a stochastic particle in a periodic potential"

_SciPost Physics, doi:SciPost Phys. 3, 018 (2017)_

## Round 1 · Referee Report · Anonymous · 2017-8-13

Strengths

1) The paper presents a reasonably clear and complete analysis of the motion of particle in a periodic potential according to a model proposed over 50 years ago by Il'in and Khas'minskii.

Weaknesses

1) The model used in the paper is not defined properly. This is especially important since the original paper introducing this model is over 50 years old and may not be easily accessible to all interested researchers.
Specifically, in the ll'in-Khas'minskii model "instantaneous interaction with the heat bath occurs at a constant rate $t_c^{-1}$". Does it mean that the interaction events are equally spaced in time? Or, are they distributed according to a probability distribution? Also, for more mathematically minded readers it would be nice to have an equation of motion for the probability distribution within the ll'in-Khas'minskii model.
2) The standard model to describe stochastic motion (the Brownian motion model) can be derived from a more fundamental (more complete) description of the particle+bath system. While the derivation is approximate, it provides some physical understanding of the assumptions behind this model. It is not clear whether (and how) the ll'in-Khas'minskii model follows from the description of the particle+bath system in terms of the coordinates, positions and interaction of the particle and the particles constituting the bath.
3) The physical interpretation and thus the estimation of the parameters in the Langevin equation describing the standard model is reasonably clear. How could one estimate $t_c$, i.e. the characteristic time of the ll'in-Khas'minskii model?
4) The paper introduces and uses a variety of acronyms. Some of them are relatively non-standard (e.g. VhD). This makes the paper a bit difficult to follow.
5) Some figures (e.g. panels (d-f) of Fig. 7) are difficult to read. The inset in panel c of Fig. 7 is impossible to read in the standard size.

Report

This paper discusses the application of an approach introduced by Il'in and Khas'minskii to the motion of a stochastic particle in a periodic potential. It provides additional details regarding previously described results. It discusses the transition from overdamped to underdamped motion. It shows that according to the Il'in-Khas'minskii model, diffusive mean-squared displacement can coexist with non-Gaussian probability distribution of displacements.
I recommend that the paper is accepted after the authors considered the (relatively minor) changes requested below.

Requested changes

1) I would appreciate some suggestions regarding the applicability of the present model and, in particular, how "the loss of a typical time ... length scale" could be accessed from the analysis of the trajectories. For example, if one had a trajectory, how would one decide whether to describe it using the present model or the Brownian motion model?
2) The model used in the paper should be properly defined.
3) The number of acronyms used should be reduced.
4) Readability of the figures should be improved.
Minor points:
a) The authors write in the Abstract that they "introduce a physically motivated theoretical approach". Rather, they use a previously introduced approach (by Il'in and Khas'minskii) to describe motion in a periodic potential.
b) According to the caption, the full line in Fig. 2a is an empirical fit rather than a prediction and thus should be labeled as such.
c) On p.7 the authors attribute the plateau to "the interaction with the thermal noise and confining barrier". I think the noise itself does not cause the plateau?
d) Some more editing (e.g. removing "Langeving" on p. 1) should be done.

---

## Round 2 · Referee Report · Anonymous (Referee 1) · 2017-8-18

Report

The authors made changes suggested in the first report.

---

## Round 2 · List of Changes

In the revised version of the manuscript :

-In Sec.2 we introduced the Il'in Kashminskii equation of motion when reviewing previous results. We specified more precisely the model

-We eliminated the acronym VhD for Van Hove, substituting with the explicit expression, and reduced the number of acronyms to improve clarity

  • We eliminated the inset in panel c of Fig 7. We have also modified panels (d-f) of Fig. 7 to improve their readability

-We modified the first sentence in the abstract and the first sentence of Sec. IIIb

-We corrected the label in Fig. 2a

-We corrected a sentence in Sec. 5a and introduced a comment at the end of Sec. 4

-We corrected Eq.20 and introduced Eq.19

-We corrected few typos

---

## Editorial Decision

published